# Bottleneck Communication Delay Minimization for Communication-Efficient Decentralized Learning

**Nozomi Hata** [1]   **Kenta Niwa** [1]

## Abstract

For communication-efficient decentralized learning, advanced network (NW) topologies, such as exponential and 1-peer exponential graphs, have been studied under homogeneous communication delays. However, real-world NWs exhibit heterogeneous communication delays, making node assignment optimization crucial for minimizing the Bottleneck Communication Delay (BCD). We propose BTSP-MSR, an approximate method for minimizing BCD on circulant digraphs, including exponential and 1-peer exponential graphs. Leveraging the fact that circulant digraphs can be viewed as a union of (directed) ring graphs, we derive an upper bound on the BCD by combining the ring-graph BCD (BTSP) with a deviation term (MSR). We then construct a solver that sequentially minimizes these two terms. Numerical experiments show that BTSP-MSR consistently reduces BCD across several circulant digraphs with large numbers of nodes. Notably, incorporating the exponential or 1-peer exponential graph enables communication-efficient decentralized learning under heterogeneous delay settings.

https://github.com/nttcslab/BTSP-MSR

## 1. Introduction

Decentralized learning is an effective approach for training large-scale model parameters using distributed computational resources interconnected via networks (NWs) (Tang et al., 2020; Lian et al., 2017). This paradigm alternates between local updates at each node and partial mixing of model parameters among sparsely connected nodes, enabling large-scale model training without centralizing data or computational resources. It is particularly suited for scenarios where a large number of computing nodes are available in the learning process (Yuan et al., 2022), and where legal restrictions (e.g., GDPR) prohibit centralized data aggregation (Zhu & Li, 2021; Truong et al., 2021).

A key research trend in decentralized learning focuses on the design of advanced NW topologies for partial mixing. Early works primarily employ simple NW topologies like ring graphs. More recent studies have explored advanced NW topologies, such as the exponential graph (Assran et al., 2019) and the 1-peer exponential graph (Ying et al., 2021). Their theoretically larger spectral gap leads to faster convergence even with simple decentralized algorithms, e.g., Decentralized SGD (DSGD) (Lopes & Sayed, 2008; Nedić et al., 2018; Koloskova et al., 2021), enabling communication-efficient decentralized learning. However, these evaluations are typically conducted under idealized NW settings that assume homogeneous communication delays across all nodes. However, in realistic physical networks (e.g., geographically distributed data centers), heterogeneous communication delays are natural (see Figure 1(b)).

Since most decentralized learning algorithms rely on synchronization across all nodes, their communication efficiency in the presence of heterogeneous communication delays is primarily constrained by the Bottleneck Communication Delay (BCD). Our target problem is to assign virtual nodes from a given NW topology (Figure 1(a)) onto a physical NW (Figure 1(b)), with the goal of minimizing the BCD under given communication delays. However, without careful node assignment–as illustrated in Figure 1(c)–the resulting NW may include node connections with large BCD. Therefore, node assignment optimization, as shown in Figure 1(d), is essential for BCD minimization.

Despite its importance, node assignment optimization is challenging: the number of possible assignments scales as $n!$ with the number of nodes $n$, making an exhaustive search impractical. Only a few studies have addressed efficient node assignment optimization under heterogeneous communication delays, and their scope remains limited. For example, Marfoq et al. (2020) studies algorithms to mini-

---

[1]NTT Communication Science Laboratory, NTT Inc., Kyoto, Japan. Correspondence to: Nozomi Hata <nozomi.hata@ntt.com>, Kenta Niwa <kenta.niwa@ntt.com>.

*Proceedings of the 43rd International Conference on Machine Learning*, Seoul, South Korea. PMLR 306, 2026. Copyright 2026 by the author(s).

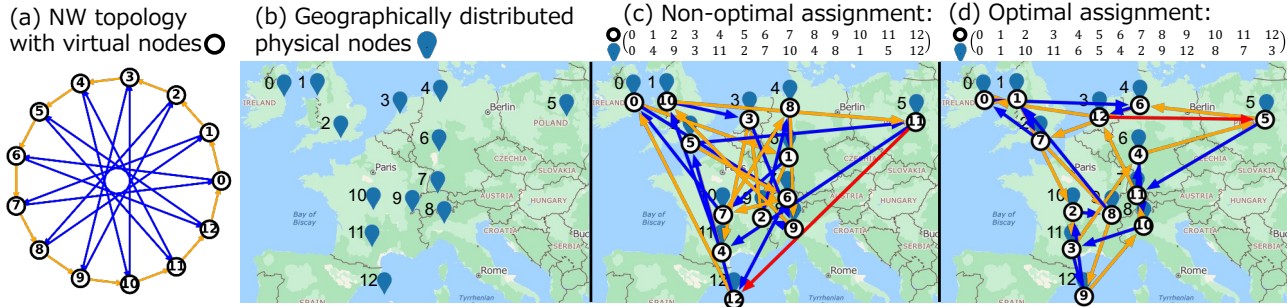

*Figure 1.* Node assignment under a given NW topology and a physical NW. (a) A target NW topology represented as a circulant digraph. (b) A physical NW consisting of geographically distributed computing nodes. (c) A non-optimal random node assignment. (d) Optimal node assignment. Assuming that communication delay depends on the physical distance between nodes, the edge representing BCD is highlighted in red. Compared with the random assignment in (c), the optimal node assignment in (d) reduces BCD, enabling communication-efficient decentralized learning.

mize average communication delay, but focuses on ring and tree graphs, as these structures are well studied and efficient solvers are available. To the best of our knowledge, node assignment optimization under heterogeneous communication delays for advanced NW topologies (e.g., exponential and 1-peer exponential graphs) remains unexplored.

In this paper, we propose BTSP-MSR, a novel approximation solver for node assignment optimization for BCD minimization under heterogeneous communication delays. We first observe that the advanced NW topologies (e.g., exponential and 1-peer exponential graphs) belong to the class of circulant digraphs. Thus, we begin by formulating the BCD problem, aiming to assign an arbitrary NW topology in a circulant digraph–to a physical NW in order to minimize BCD (see Sec. 2). To approximately solve this problem, we aim to represent the BCD upper bound by using tractable terms by exploiting the structural properties of circulant digraphs. Specifically, since any circulant digraph can be viewed as a union of (directed) ring graphs, the BCD upper bound can be represented by (i) the ring-graph BCD term (Bottleneck Traveling Salesman Problem: BTSP) and (ii) a deviation term (Maximum Skip-sum to be Reduced: MSR) capturing the gap between the target circulant digraph and the ring graph. Sequentially minimizing these tractable terms yields an approximately optimal node assignment.

Our contributions are summarized as follows:

**Approximation Solver for BCD on Circulant Digraph Using Tractable Terms (Sec. 4.1–4.2).** Leveraging the fact that any circulant digraph can be viewed as a union of ring graphs, we derive an upper bound of BCD in terms of two tractable terms: (i) BTSP term and (ii) MSR term (see Theorem 1). Based on this, we construct BTSP-MSR, which sequentially minimizes two tractable terms in our derived BCD upper bound (see Algorithm 1).

**Theoretical Analyses of BTSP-MSR (Sec. 4.3).** We provide theoretical approximation ratios of BTSP-MSR in two scenarios. After establishing the worst-case approximation

ratio in Theorem 2, we derive a tighter approximation ratio under certain assumptions motivated by practical NW scenarios in Theorem 3.

**Experimental Validations (Sec. 5).** Our experiments show that pairing BTSP-MSR with advanced NW topologies (exponential and 1-peer exponential graphs) maximizes communication efficiency in decentralized learning by jointly (i) reducing communication delays through BTSP-MSR and (ii) leveraging the strong spectral gap of these topologies.

## 2. Problem Formulation

**(a) Decentralized Learning.** In decentralized learning, model parameters across $n$ nodes are updated synchronously by iteratively performing (i) local updates within each node $i \in \{0, \cdots, n-1\}$ and (ii) partial mixing among connected nodes under a given NW topology. The simplest decentralized learning algorithm, DSGD (Lopes & Sayed, 2008), updates the model parameters across $n$ nodes–represented at each communication round $l \in \{0, \ldots, R-1\}$ by $X^{(l)} = (\boldsymbol{x}_0^{(l)}, \ldots, \boldsymbol{x}_{n-1}^{(l)}) \in \mathbb{R}^{m \times n}$ as follows:

**(i) Local update** $\quad X^{(l+1/2)} = X^{(l)} - \eta \nabla F(X^{(l)})$,

**(ii) Partial mixing** $\quad X^{(l+1)} = X^{(l+1/2)} W^{(\mathrm{mod}\ (l,\tau))}$, $\quad$ (1)

where $\eta$ denotes a learning rate, $\nabla F(X^{(l)}) = (\nabla f_0^{(l)}(\boldsymbol{x}_0^{(l)}), \ldots, \nabla f_{n-1}^{(l)}(\boldsymbol{x}_{n-1}^{(l)})) \in \mathbb{R}^{m \times n}$ is stochastic local gradients, $W^{(l)} \in \mathbb{R}_+^{n \times n}$ is mixing matrix. For partial averaging, the entry of the mixing matrix $W_{ij}^{(l)}$ is positive if nodes $i$ and $j$ are connected, and zero otherwise. The $W^{(l)}$ is typically designed to be sparse–reducing communication overhead and reflecting the underlying NW topology. It is predefined before training and remains fixed throughout the decentralized learning process (see Appendix B). Our notation supports both static ($\tau = 1$) and dynamic NW topologies, where the dynamic case is represented by a cyclic sequence of $\tau (\geq 2)$ mixing matrices $\{W^{(0)}, \ldots, W^{(\tau-1)}\} \in \mathbb{R}_+^{(n \times n) \times \tau}$.

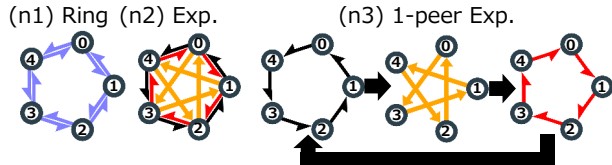

*Figure 2.* NW topologies represented as circulant digraphs: (n1) ring, (n2) exponential, and (n3) 1-peer exponential graphs with $n = 5$. The 1-peer exponential graph is a dynamic graph that decomposes the exponential graph into a sequence of 1-peer communications. It cyclically applies $\tau = \lceil \log_2 n \rceil (\lceil \log_2 5 \rceil = 3)$ distinct mixing matrices.

A convergence rate of DSGD in (1) is shown in (Koloskova et al., 2021), which depends on the spectral gap of the mixing matrix[1]. While additional details are provided in Appendix B, it is well established that employing NW topologies with larger spectral gaps enables more communication-efficient decentralized learning by accelerating the convergence of model parameters.

For communication-efficient decentralized learning, it is advantageous to employ advanced NW topologies that enlarge the spectral gap, rather than relying on simpler ones such as the ring graph. A prominent example is the exponential graph (Assran et al., 2019), in which each node connects to others at exponentially skipped intervals, as illustrated in Figure 2(b). This graph offers a larger spectral gap by effectively balancing short- and long-skip connections. An extension of this is the 1-peer exponential graph (Ying et al., 2021), which decomposes the connections in the (static) exponential graph to a dynamic one-peer communications across multiple communication rounds.

We notice that these advanced NW topologies (exponential and 1-peer exponential graphs) and the commonly used ring graph, can all be represented as circulant digraphs:

**Definition 1** (Circulant Digraph). A square matrix $A \in \mathbb{R}^{n \times n}$ is called a *circulant matrix* if there exists a vector $(a_0, \ldots, a_{n-1}) \in \mathbb{R}^n$ such that $A_{ij} = a_{[j-i]}$, where $[i] = \mathrm{mod}(i, n)$. A directed graph $G = (V, E)$ is called a *circulant digraph* if $G$ has a circulant adjacency matrix.

However, these advanced NW topologies–such as the exponential and 1-peer exponential graphs–have predominantly been evaluated under the assumption of homogeneous communication delays, where communication delays between nodes are either ignored or assumed to be uniform.

In contrast, consider real-world NW scenarios in which physical nodes (e.g., computing servers in multiple data centers) are geographically distributed, as illustrated in Figure

1(b), and are interconnected via physical edges (e.g., optical fibers and NW switches). In such settings, it is natural to assume that communication delays across edges are heterogeneous. To implement a specific NW topology–such as ring, exponential, or 1-peer exponential graph–within real-world NW scenarios, it is necessary to assign virtual nodes from a given NW topology to physical nodes in the underlying infrastructure. However, without a carefully designed assignment, communication bottlenecks may arise, leading to significant inefficiencies in decentralized learning under real-world NW scenarios (Figure 1 (c)).

**(b) Bottleneck Communication Delay (BCD).** The optimization problem of interest for communication-efficient decentralized learning is formulated. Given a specific NW topology, such as ring, exponential, or 1-peer exponential graph, our goal is to assign virtual nodes from the given NW topology to physical nodes in the underlying infrastructure in order to minimize the Bottleneck Communication Delay (BCD) before starting decentralized learning. In particular, we focus on circulant digraphs, including ring, exponential, and 1-peer exponential graphs, due to their practical relevance and desirable spectral gap properties. To formalize the problem, we introduce the following notations: $\pi(i)$ denotes the index of physical node assigned to virtual node $i$, and the communication delay between physical nodes $\pi(i)$ and $\pi(j)$ for each $(i, j) \in E^{(l)}$ is denoted by $d\left(\pi(i), \pi(j)\right)$. Since $d$ is determined by the underlying physical NW characteristics, it is given and uncontrollable. For practical deployment, we therefore may need to account for worst-case realizations of $d$. In this paper, we assume that $d$ is a (static) distance function; that is, the communication delays between two sites are symmetric and time-invariant[2].

For dynamic NW topologies–such as the 1-peer exponential graph illustrated in Figure 2(c)–which consist of $\tau(\geq 2)$ mixing matrices, the goal is to find an assignment $\pi$ that minimizes the BCD averaged over $\tau$ communication rounds:

$$\min_{\pi} \left( \varphi(\pi; \mathcal{E}) = \frac{1}{\tau} \sum_{l=0}^{\tau-1} \max_{(i,j) \in E^{(l)}} d\left(\pi(i), \pi(j)\right) \right), \quad (2)$$

where $\mathcal{E} = \{E^{(0)}, \ldots, E^{(\tau-1)}\}$ denotes the edge sets over $\tau$ communication rounds, and $\tau = 1$ is available for static NW topologies (e.g., ring and exponential graphs).

Next, we explain that minimizing the BCD in (2) is NP-hard, indicating computational intractability:

**Proposition 1.** *Minimizing the BCD is NP-hard.*

*Proof.* Consider the case of the ring graph with $\tau = 1$. In this setting, the edge sets connect cyclically neighboring

---

[1]For dynamic NW topologies, the spectral gap is defined in expectation, referring to the spectral gap of the expected product of mixing matrices applied over multiple communication rounds, and its formal definition is given in Assumption 3 in Appendix B.

[2]We empirically investigate the impact of non-distance-based communication delays by evaluating robustness to time-varying asymmetrical communication delays in Appendix F.6.

nodes as $E^{(0)} = \{(0,1), (1,2), \cdots, (n-1,0)\} \cup \{(0, n-1), (n-1, n-2), \cdots, (1,0)\}$. Since $d$ is a distance function–that is, it satisfies symmetry–the BCD objective $\varphi(\pi, \mathcal{E})$ reduces to that of the BTSP, which is known to be NP-hard (Parker & Rardin, 1984). Thus, minimizing the BCD is also NP-hard since it includes BTSP as a specific case. $\square$

Since minimizing the BCD is NP-hard even under simple NW topologies–such as ring graph–employing more advanced NW topologies, such as exponential graphs and 1-peer exponential graphs, introduces additional complexity, especially for large $n$. Thus, we aim to construct an approximation solver for the BCD tailored to circulant digraphs.

## 3. Related Works

**(a) BTSP Approximation Solver.** When using the (directed) ring graph, the BCD objective (2) reduces to the BTSP objective. Several approximation solvers for BTSP are known (Parker & Rardin, 1984; Kao & Sanghi, 2009; An et al., 2021), and in particular, a 2-approximation solver is available with a distance function $d$. Its computational complexity is $O(n^2 \log n)$, making it practical for moderately large $n$. However, such approximation solvers are not applicable to more advanced NW topologies (e.g., exponential and 1-peer exponential graphs).

**(b) Minimum Cycle Time Problem (MCT) Solvers.** In (Marfoq et al., 2020), solvers for the Minimum Cycle Time problem (MCT) are studied by leveraging the fact that the average communication delay can be efficiently computed for certain NW topologies, such as tree and ring graphs. Specifically, (i) Prim's algorithm is employed with $O(n^2)$ complexity to obtain an exact solution for tree graphs, and (ii) Christofides' algorithm is used with $O(n^2 \log n)$ complexity to yield a $3n$-approximate solution for the ring graph, both of which offer practical solutions. However, they are applicable only to these limited NW topologies. To the best of our knowledge, no prior work has addressed solving the BCD problem under advanced NW topologies (e.g., exponential and 1-peer exponential graphs).

## 4. Proposed Method

We propose BTSP-MSR, an approximation solver for BCD that is applicable to any circulant digraphs, including ring, exponential, and 1-peer exponential graphs. Our key approach is motivated by the observation that any circulant digraph can be viewed as a union of (directed) ring graphs. For example, as shown in Figure 3, the exponential graph with $n=5$ nodes can be viewed as a union of three (directed) ring graphs. Building on this observation, in Sec. 4.1, we reformulate the BCD objective (2) for circulant digraphs and derive an upper bound consisting of two tractable terms: (i)

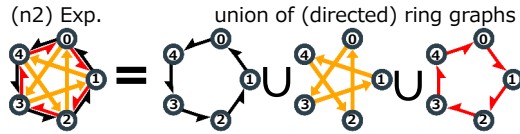

(n2) Exp.          union of (directed) ring graphs

*Figure 3.* A circulant digraph (e.g., exponential graph) can be represented as a union of (directed) ring graphs.

BTSP term, corresponding to the BCD on a ring graph and (ii) MSR term, which captures the deviation between the target circulant digraph and the ring graph and is minimized to reduce this mismatch. In Sec. 4.2, we develop an approximation solver to minimize this upper bound, which we refer to as BTSP-MSR (Algorithm 1). In Sec. 4.3, we investigate approximation ratios of BTSP-MSR in two scenarios: a worst-case scenario and a practically NW motivated one.

### 4.1. BCD Upper Bound for Circulant Digraphs

This subsection consists of three parts: (a) we begin by reformulating the BCD for circulant digraphs (Lemma 1); (b) we derive an upper bound using two tractable terms–the BTSP and MSR terms (Lemma 2); and (c) we further refine the MSR term to obtain a tighter bound in Theorem 1.

**(a) BCD Reformulation for Circulant Digraphs.** We begin by reformulating (2) to incorporate the structural properties of circulant digraphs. Owing to their underlying cyclic structure, the edge set $E^{(l)}$ in a circulant digraph can be expressed using a concise notation. For this aim, we introduce the edge generator $S^{(l)}$, defined as follows:

**Definition 2** (Edge Generator for Circulant Digraphs). Let $E^{(l)}$ be the edge set of a circulant digraph with $n$ nodes. Then the edge generator $S^{(l)}$ of $E^{(l)}$ is defined as $S^{(l)} = \{i \in \{1, \ldots, n-1\} \mid (0, i) \in E^{(l)}\}$.

Using the edge generator $S^{(l)}$, the edge set can be represented as $E^{(l)} = \{(i, j) \mid \mod(j - i, n) \in S^{(l)}\}$. For example, the edge generator for the exponential graph shown in Figure 2(b) is given by $S^{(0)} = \{1, 2, \ldots, 2^{\lceil \log_2 n \rceil - 1}\}$, while that for the 1-peer exponential graph in Figure 2(c) is $S^{(l)} = \{2^l\}$, where $\{0, \ldots, \tau-1\} \in l$ and $\tau = \lceil \log_2 n \rceil$. Each element $s \in S$ represents a connection interval that skips over nodes from a base node, which we refer to as a skip width. Using this notation, BCD in (2) can be reformulated to fit circulant digraphs:

**Lemma 1** (BCD Reformulation for Circulant Digraphs). *Let $\mathcal{S} = (S^{(0)}, \ldots, S^{(\tau-1)})$ be edge generators regarding $\mathcal{E} = (E^{(0)}, \ldots, E^{(\tau-1)})$ of $\tau$ circulant digraphs, respectively. Then, we can rewrite (2) as*

$$\min_{\pi} \left( \varphi(\pi; \mathcal{E}) = \frac{1}{\tau} \sum_{l=0}^{\tau-1} \max_{i \in V, s \in S^{(l)}} d(\pi(i), \pi([i+s])) \right), (3)$$

*where $[i] = \mod(i, n)$.*

**(b) BCD Upper Bound Using Tractable Terms.** To approximately solve (3), we derive an upper bound composed of tractable terms: (i) BTSP term and (ii) MSR term. As noted earlier, our approach leverages the fact that a circulant digraph can be viewed as a union of (directed) ring graphs, which motivates searching around a BTSP solution.

First, we introduce a notation to represent the BTSP objective. Since the BTSP is equivalent to minimizing the BCD on the ring graph, its objective $B(\pi)$ can be obtained by substituting the ring graph parameters–specifically, $S^{(0)}=\{1\}$ and $\tau = 1$–into the objective in (3): that is,

$$B(\pi) = \max_{i \in V} d(\pi(i), \pi([i+1])). \tag{4}$$

As noted in Sec. 3, a 2-approximation solution for the problem $\min_\pi B(\pi)$ can be obtained with the BTSP approximation solver (Parker & Rardin, 1984).

Second, we derive an inequality involving $d(\pi(i), \pi([i+s]))$ in (3), to relate it to the BTSP objective $B(\pi)$. Since $d$ is assumed to be a distance function, direct communication between two physical nodes is always no slower than communication through an intermediate physical node, the following triangle inequality holds:

$$d(\pi(i), \pi([i+s])) \leq d(\pi(i), \pi([i+1])) + d(\pi([i+1]), \pi([i+s])),$$

where $d(\pi(i), \pi([i+1]))$ is used in the RHS since it appears in the BTSP objective (4). By recurrently repeating this for $s$ times and using the definition of $B(\pi)$, we have

$$d(\pi(i), \pi([i+s])) \leq \sum_{k=0}^{s-1} d(\pi([i+k]), \pi([i+k+1])) \leq sB(\pi),$$

which provides an upper bound that captures the cumulative effect of long skip connections ($s>1$). Using this inequality, the objective in (3) is bounded using BTSP objective $B(\pi)$:

**Lemma 2** (BCD Upper Bound Using Tractable Terms). *Let $\mathcal{E}=(E^{(0)}, \cdots, E^{(\tau-1)})$ be edge sets generated from $\mathcal{S}=(S^{(0)}, \cdots, S^{(\tau-1)})$, respectively. Then, we have*

$$\varphi(\pi; \mathcal{E}) \leq B(\pi)g(\mathcal{S}), \tag{5}$$

*where $g(\mathcal{S}) = \frac{1}{\tau}\sum_{l=0}^{\tau-1} \max S^{(l)}$ denotes the Maximum Skip-sum to be Reduced (MSR) term.*

When the underlying NW topology is the (directed) ring graph, $g(\mathcal{S}) = 1$, and the BCD reduces to the BTSP. In contrast, multiple skip widths in the NW topology cause the trade-off, which leads to $g(\mathcal{S}) > 1$. Hence, the MSR term $g(\mathcal{S})$ can be regarded as a deviation from the given circulant digraph and the (directed) ring graph.

While a 2-approximate solution $\pi_1$ can be obtained for the BTSP (i.e., $\min_\pi B(\pi)$), this yields a $2g(\mathcal{S})$-approximate solution depending on the MSR term; thus, we further explore minimizing $g(\mathcal{S})$ for a tighter BCD upper bound.

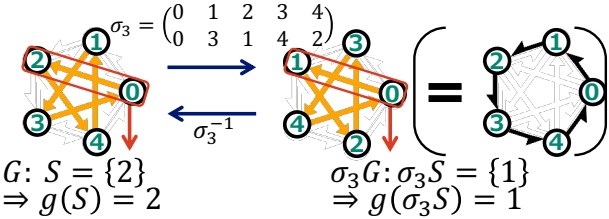

$G: S = \{2\}$
$\Rightarrow g(S) = 2$

$\sigma_3 G: \sigma_3 S = \{1\}$
$\Rightarrow g(\sigma_3 S) = 1$

*Figure 4.* Two circulant digraphs $\{G, \sigma_3 G\}$, that share the same underlying NW topology but differ in their edge generators. A suitable node permutation reduces the maximum skip-sum.

**(c) Further Reduction of MSR Term.** To obtain a tighter BCD upper bound, we introduce a method to minimize the MSR term. Recall that $g(\mathcal{S})$ is governed by $\max S^{(l)}$, which tends to increase with the inclusion of long-skip connections. Thus, $g(\mathcal{S})$ can vary depending on the ordering of virtual nodes, even when the underlying NW topology remains unchanged. To illustrate this, Figure 4 presents an example of two circulant digraphs that share the same NW topology but differ in their employing edge generators. The LHS depicts a graph $G=(V, E)$ with $n$ nodes, where the maximum skip-sum $g(\mathcal{S})$ is relatively large due to the presence of long-skip connections. In contrast, the RHS shows a permuted graph $\sigma G = (V, \sigma E)$, where $\sigma E = \{(\sigma(i), \sigma(j))|(i, j) \in E\}$ is permuted edge set using a node permutation $\sigma : V \to V$. To ensure that $\sigma$ preserves the circulant structure (Dobson & Morris, 2002), we restrict $\sigma$ to a multiplicative permutation (Ádám, 1981), denoted by $\sigma_c(i) = [ci]$, where $c (\leq n)$ is an integer that is coprime to $n$. By choosing the permutation to eliminate long-skip connections, the MSR term can be minimized.

With the introduction of the multiplicative permutation $\sigma_c$, the BCD upper bound inequality (5) can be refined:

**Theorem 1** (BCD Upper Bound Using Multiplicative Permutation). *Let $c$ be a positive integer coprime to the number of nodes $n$. For a multiplicative permutation $\sigma_c$, we have*

$$\varphi(\pi \circ \sigma_c; \mathcal{E}) \leq B(\pi)g(\sigma_c \mathcal{S}), \tag{6}$$

*where $\sigma_c \mathcal{S} = (\sigma_c S^{(0)}, \cdots, \sigma_c S^{(\tau-1)})$ and $\sigma_c S^{(l)} = \{[cs^{(l)}] \mid s^{(l)} \in S^{(l)}\}$ for $l \in \{0, \ldots, \tau-1\}$.*

Since the first term on the RHS, $B(\pi)$, remains in (5), the upper bound can be reduced by searching for $c$ that reduces $g(\sigma_c \mathcal{S})$; that is, $\min_{\pi,c} B(\pi)g(\sigma_c \mathcal{S}) \leq \min_\pi B(\pi)g(\mathcal{S})$. Moreover, minimizing the MSR term also improves the achieved BCD itself; for a BTSP solution $\pi_1$, we have $\min_c \varphi(\pi_1 \circ \sigma_c; \mathcal{E}) \leq \varphi(\pi_1; \mathcal{E})$. As $c$ is restricted to integers less than $n$ and coprime to $n$, it can be identified in $O(n)$ time. Furthermore, using the symmetry of $d$, we can reduce $g(\sigma_c \mathcal{S}) \leq \lfloor n/2 \rfloor$ (see Lemma 3 in Appendix C).

## 4.2. BTSP-MSR for BCD Upper Bound Minimization

Based on Theorem 1, we aim to solve the minimization of the upper bound: $\min_{\pi,\sigma_c} B(\pi)\, g(\sigma_c\mathcal{S})$ by sequentially solving two subproblems: minimizing $B(\pi)$ and $g(\sigma_c\mathcal{S})$. For this aim, we construct BTSP-MSR, a two-stage solver to approximately minimize the BCD on circulant digraphs, as described in Algorithm 1. In the 1st stage, it computes an initial solution $\pi_1$ via a 2-approximation for BTSP (Parker & Rardin, 1984): $\min_\pi B(\pi)$. In the 2nd stage, the solution is refined by minimizing MSR term $g(\sigma_c\mathcal{S})$ to improve the approximation ratio.

Next, the computation cost for BTSP-MSR is investigated. The 1st stage requires $O(n^2 \log n)$ time (Parker & Rardin, 1984), while the 2nd stage requires $O(n)$; therefore, the overall time complexity of BTSP-MSR remains $O(n^2 \log n)$. This complexity is efficient even for large $n$; BTSP-MSR completes within a few minutes for $n \leq 2,000$ in the experiment (see Appendix E). Since node assignment is performed only once before the decentralized learning, its computational overhead is negligible compared to the total training cost, indicating the practicality of BTSP-MSR.

## 4.3. Approximation Ratio Analyses for BTSP-MSR

The approximation ratio of BTSP-MSR is investigated. First, for an arbitrary NW topology and distance function $d$ to represent communication delays among $n$ nodes, we get the following approximation ratio.

**Theorem 2** (Approximation Ratio of BTSP-MSR (worst–case bound)). *Assume that the optimal BCD is larger than the optimal BTSP objective[3]: $\min_\pi B(\pi) \leq \min_\pi \varphi(\pi; \mathcal{E})$. Then, by minimizing BCD with BTSP-MSR (Algorithm 1), the solution $\pi^*$ achieves an approximation ratio of $2\min_c g(\sigma_c\mathcal{S})$; that is, $\varphi(\pi^*; \mathcal{E}) \leq 2 \min_c g(\sigma_c\mathcal{S}) \min_\pi \varphi(\pi; \mathcal{E})$.*

The proof is provided in Appendix D. Theorem 2 indicates that the approximation ratio is primarily governed by the MSR term (and its minimization via $\sigma_c$). Because the MSR term depends on the underlying NW topology, one can obtain a tight bound for specific NW topologies–especially those that include long-skip connections for small values of $c$[4]. On the other hand, for NW topologies that cannot reduce $g(\sigma_c\mathcal{S})$, the bound in Theorem 2 may not be tight.

However, under practical NW scenarios such that the following assumptions hold, we can derive a tighter bound on the approximation ratio. In practical NW scenarios, physical nodes are typically geographically distributed in a roughly random manner. Moreover, Gueye et al. (2004) suggests a

---

[3]This assumption holds in most circulant digraphs, including exponential graph, detailed in Appendix D.

[4]We additionally evaluate (n4) sparse exponential graph, including long-skip connections for small values of $c$, in Sec. 5.

---

**Algorithm 1** BTSP-MSR to Solve BCD on Circulant Digraph

1: **Input:** Number of nodes $n$, communication delay function capturing physical NW properties $d = (d_{pq}) \in \mathbb{R}^{n \times n}$, edge set of NW topology (circulant digraph) $\mathcal{E} = (E^{(0)}, \cdots, E^{(\tau-1)})$
2: **Output** $\pi^*$: $2\min_c g(\sigma_c\mathcal{S})$-approximate BCD solution to assign an input NW topology to a physical NW
3: **//(1st stage) BTSP approximation.**
4: $\pi_1 \leftarrow$ BTSP-ApproximationSolver($d$)
5: $\pi^* \leftarrow \pi_1$
6: **//(2nd stage) MSR using multiplicative permutation.**
7: **for** $c = 1, \cdots, n-1$ s.t. $c$ is coprime to $n$ **do**
8:     **if** $\varphi(\pi_1 \circ \sigma_c; \mathcal{E}) < \varphi(\pi^*; \mathcal{E})$ **then**
9:         $\pi^* \leftarrow \pi_1 \circ \sigma_c$
10:     **end if**
11: **end for**
12: **Return** $\pi^*$

---

simple delay model in which the communication delay between physical nodes can be approximated by the Euclidean distance of the communication path, up to a small additive overhead (bias). Moreover, physical nodes are randomly distributed, making the BTSP route unlikely to be highly aligned. Then, the communication directions along the route tend to be weakly correlated. To capture these characteristics, we introduce the following two assumptions.

**Assumption 1** ((Informal) Euclidean Distance-based $d$)**.** The distance function $d$ to represent communication delay is a linear function of the Euclidean distance between physical nodes with a communication overhead bias $b(\geq 0)$.

**Assumption 2** ((Informal) Bounded Maximum Edge-direction Correlation)**.** There exists a constant $\beta(\geq 0)$ such that the maximum edge-direction correlation $h(s)$ (specified in Appendix D) is bounded by $\beta$ with high probability.

**Theorem 3** ((Informal) Approximation Ratio of BTSP-MSR under Practical NW Scenarios)**.** *Suppose that Assumptions 1 and 2 hold. Then, the solution of BTSP-MSR $\pi^*$ satisfies $\varphi(\pi^*; \mathcal{E}) \leq 2 \min_c \sqrt{g(\sigma_c\mathcal{S}) + \beta(\min_\pi \varphi(\pi; \mathcal{E}) - b)} + b$ with high probability.*

The formal statements and proof are provided in Appendix D. Comparing Theorems 2 and 3, the dependence on the MSR term is reduced to a square-root factor; consequently, the adverse impact of unfavorable NW topologies is mitigated. Moreover, for the constant $\beta$ in Assumption 2, we empirically observed that $\beta$ is not large in certain NW scenarios (see Figure 16 in Appendix E.5). In summary, our approximation ratio analyses indicate that BTSP-MSR (Algorithm 1) achieves high-quality approximations for circulant digraphs under Assumptions 1 and 2, which are expected to hold in practical NW scenarios.

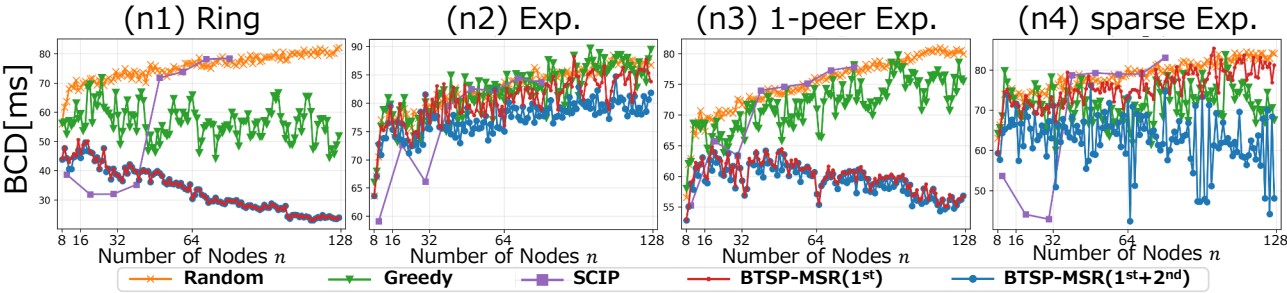

*Figure 5.* Evaluation of BCD under four NW topologies across a wide range of nodes $n$: (n1) ring, (n2) exponential, (n3) 1-peer exponential, and (n4) sparse exponential graphs. Due to computational resource limitations, SCIP was evaluated only for a subset of node sizes $n$. For large $n$, SCIP failed to obtain any solutions within the time limit. Our BTSP-MSR outperforms baselines in nearly all instances, with the performance gap becoming more highlighted as the number of nodes increases.

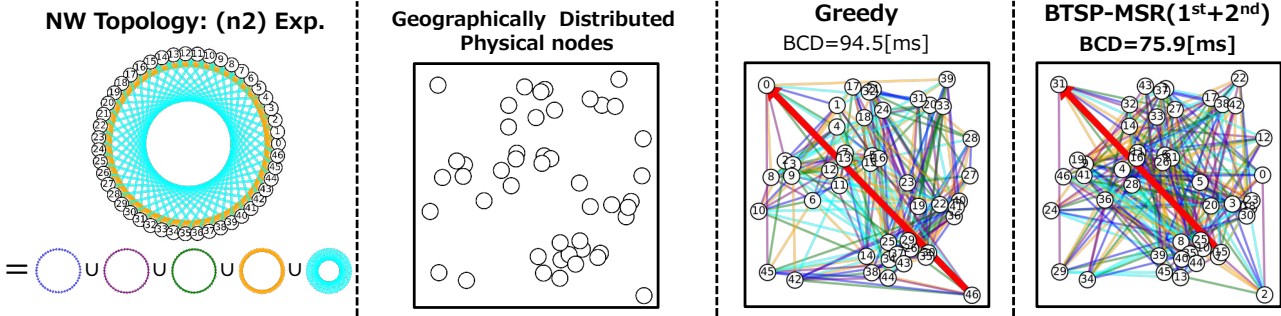

*Figure 6.* Visualization of node assignments for the (n2) exponential graph with $n = 47$ under experiment (T1), along with the solutions of the BCD obtained by the greedy solver and BTSP-MSR(1$^{st}$+2$^{nd}$). The physical nodes are randomly placed within a square. Each red edge indicates the BCD under the given NW topology and assignment.

## 5. Experiments

To demonstrate the effectiveness of BTSP-MSR, we conducted two experiments: (T1) evaluated the BCD using several circulant digraphs across a wide range of $n$; and (T2) investigated the BCD and its impact on decentralized learning under a real-world physical NW scenario.

**(T1) BCD Evaluation Under Simulated Physical NWs.**

**Physical NWs.** To investigate BCD behavior over several circulant digraphs across a wide range of network size $n \in [8, 128]$, we simulated geographically embedded networks by placing $n$ nodes uniformly at random over an $8,000 \times 8,000$ km square.[5] Following the experiments in (Marfoq et al., 2020), the communication delay of two nodes $p$ and $q$ was set as distance: $d(p, q) = 4 + 0.0085 \times \mathrm{dis}(p, q)$[ms], where $\mathrm{dis}(p, q)$ is the Euclidean distance in (T1). (We additionally investigate robustness to time-varying and asymmetrical communication delays in Appendix F.6.) For each $n$, we evaluated five different random seeds and reported the average performance.

**NW Topologies.** We prepare four NW topologies, each represented by a circulant digraph: (n1) ring graph

($|S|=2, \tau=1$), (n2) exponential graph ($|S|=\lceil \log_2 n \rceil, \tau = 1$), (n3) 1-peer exponential graph ($|S|=1, \tau=\lceil \log_2 n \rceil$), and (n4) sparse exponential graph ($|S|=2, \tau=1$), obtained by retaining only the shortest and longest connections from the exponential graph. As discussed below Theorem 2, sparser NW topologies tend to yield smaller approximation ratios; accordingly, we additionally include (n4) to empirically assess the impact of edge sparsity.

**Baselines.** We consider three baselines: *Random*: a random node assignment as the simplest baseline, *Greedy*: a heuristic solver based on a greedy strategy (e.g., (Papadimitriou & Steiglitz, 1998)) which incrementally builds a solution by locally selecting assignments that minimize the BCD, and *SCIP* (Achterberg, 2009): an open-source exact solver for combinatorial optimization problems, which integrates local search heuristics while guaranteeing optimality. The greedy solver offers computational efficiency but lacks performance guarantees and is susceptible to getting trapped in local optima. In contrast, SCIP can search for a globally optimal solution; however, it requires exponential computation time, especially with a large number of nodes $n$. We therefore impose a 30-minute time limit on SCIP, which also underscores its impracticality for large-scale instances.

In addition, we evaluate *BTSP-MSR* using Algorithm 1. We denote the full method as *BTSP-MSR (1$^{st}$+2$^{nd}$)* and its

---

[5]In (T2), we use node position information from a real-world physical NW (Ebone NW (Mahajan et al., 2002)) spanning Europe and the United States. To match its geographic scale, we rescale the synthetic node placements to this domain.

ablation that utilizes only 1st stage as *BTSP-MSR (1st)*. The latter serves as a ring-graph-specific baseline. We evaluate each solver over five trials and report the average score.

**Results.** Figure 5 shows the BCD for the four NW topologies over a wide range of $n$. As $n$ increases, typical nearest-neighbor distances decrease, leading to lower BCD for the ring graph. By contrast, for the other NW topologies in Figure 5, the presence of long-skip connections prevents a comparable reduction in BCD. BTSP-MSR $(1^{st}+2^{nd})$ out-performs all baselines in nearly all instances, with the performance gap becoming more highlighted as $n$ increases–particularly for the (n1) ring and (n3) 1-peer exponential graphs. This is expected because these graphs are dominated by short-skip connections, where improved assignments can substantially reduce BCD. Comparing BTSP-MSR $(1^{st}+2^{nd})$ with its ablation highlights the contribution of the MSR $(2^{nd}$ stage in BTSP-MSR): MSR is especially effective on (n2) exponential and (n4) sparse exponential graphs, which include long-skip connections in every communication round. This is consistent with our earlier discussion that the MSR term can be sufficiently reduced in certain sparse circulant digraphs with long-skip connections. For greedy solver and SCIP, their solution quality deteriorates as the number of nodes $n$ increases. This is likely because the greedy solver tends to get trapped in local optima, while SCIP cannot explore an exponentially large solution space within the 30-minute time limit; for large $n$, SCIP even failed to return any solutions within the time limit. Appendix E reports the detailed results. In summary, the experiment (T1) empirically confirms that BTSP-MSR contributes to BCD reduction across diverse NW topologies and physical NWs.

Figure 6 illustrates an example with $n = 47$ nodes. The greedy solver fails to avoid long-range communication due to local optimization, whereas BTSP-MSR$(1^{st}+2^{nd})$ avoids this by considering all nodes and skip widths, reducing the BCD by approximately 20%. Appendix E includes additional node assignment examples with other NW topologies.

**(T2) BCD and Decentralized Learning Evaluations Under a Real-world Physical NW.**

**Physical NW.** To emulate communication delays in real-world physical NWs, we use the Ebone NW (Mahajan et al., 2002), consisting of $n = 87$ nodes spanning Europe and the United States. We compute the communication delay between two nodes $p$ and $q$ using $d(p,q)$ as in (T1). However, since $p$ and $q$ are not always directly connected, we use their shortest-path distance as $\text{dis}(p,q)$. The Ebone NW and the results are visualized in Appendix F.7.

**NW Topologies.** In addition to four circulant digraphs (n1)-(n4), we include MATCHA+ (Wang et al., 2022), which is not a circulant digraph. MATCHA+ randomly connects with a preference for geographically neighboring nodes.

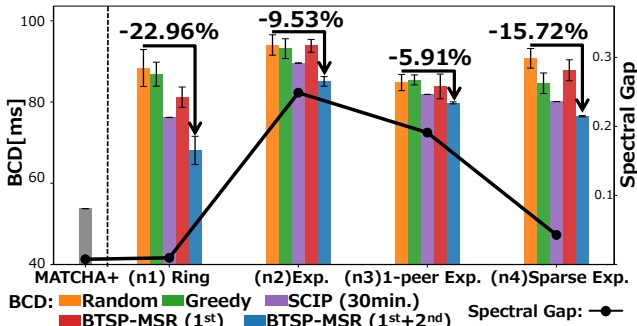

*Figure 7.* BCDs (in milliseconds) averaged over five independent runs and estimated spectral gaps for several NW topologies with Ebone NW ($n = 87$). The error bars (black lines) indicate a standard deviation of BCDs. For circulant digraphs, we additionally reported the speed-up of BCDs relative to random assignment at the top of the figure.

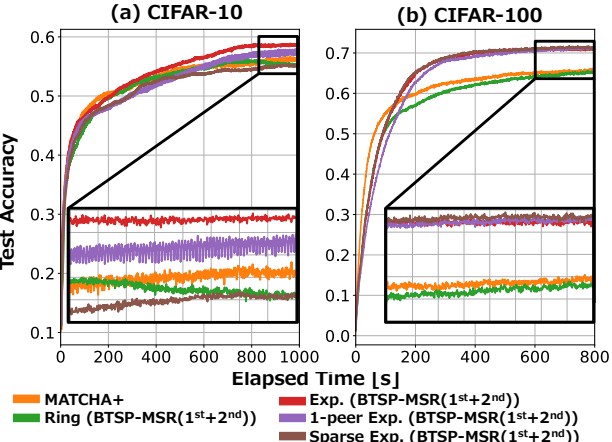

*Figure 8.* Decentralized learning results on the (a) CIFAR-10 and (b) CIFAR-100 datasets. Accounting for communication delays, the horizontal axis represents elapsed time.

**Datasets and Model for Decentralized Learning.** In (T2), We train LeNet-5 (LeCun et al., 2002) for (a) CIFAR-10 classification task and ResNet-18 (He et al., 2016) for (b) CIFAR-100 classification task. Each dataset is partitioned across $n$=87 nodes to simulate non-IID data allocation (Vogels et al., 2021), following standard practices in decentralized learning evaluations. As illustrated in Appendix F.1, strongly non-IID data settings (Dir. $\alpha$=0.1) result in varying numbers of data samples across nodes.

**Results.** Figure 7 reports the BCD results and estimated spectral gaps on the Ebone NW ($n$=87). We initialize SCIP with the identity permutation, since otherwise it may fail to return a solution for large $n$. As shown in Figure 7, across all tested circulant digraphs, BTSP-MSR consistently achieves lower BCDs than all other solvers. These results largely follow the trends observed in (T1), except for the (n1) ring graph. In particular, the benefit of MSR is prominent in sparse circulant digraphs. For the ring graph, BTSP-MSR $(1^{st})$ can yield suboptimal assignments due to the approx-

imate nature of the BTSP solver; nonetheless, MSR still reduces BCD. Regarding the spectral gap, these topologies exhibit significantly larger values than the other NW topologies shown in Figure 7. These large spectral gaps, together with the BCD reduction achieved by BTSP-MSR, are expected to be effective for decentralized learning. In contrast, although MATCHA+ achieves the lowest delays by prioritizing geographically neighboring communication, the spectral gap is limited, which may limit its effectiveness for decentralized learning despite the reduced BCD.

Next, Figure 8 presents decentralized learning results using the global model parameter (i.e., the average of $n$ local models) on the two datasets, where the horizontal axis shows elapsed time accounting for communication delays. In both cases, the exponential and 1-peer exponential graphs yield higher test accuracy, reflecting their large spectral gap and efficient averaging. This demonstrates that BTSP-MSR($1^{st}$+$2^{nd}$) enables these graphs to simultaneously achieve lower BCDs and large spectral gaps. In contrast, MATCHA+ performs better in early iterations but ultimately converges to lower accuracy–likely due to the limited capacity for global information exchange. Overall, BTSP-MSR remains effective in decentralized learning tasks.

## 6. Conclusion

We propose BTSP-MSR, an approximation solver for minimizing BCD on circulant digraphs, toward communication-efficient decentralized learning. First, leveraging the fact that circulant digraphs can be viewed as a union of (directed) ring graphs, we derive a BCD upper bound composed of two tractable terms: (i) BTSP and (ii) MSR. To minimize this upper bound, we construct BTSP-MSR. Second, its approximation ratio is theoretically analyzed. Particularly, a tight approximation ratio is obtained under certain assumptions reflecting practical NW scenarios. Third, through numerical experiments, we showed that BTSP-MSR consistently reduces BCD. Notably, for the exponential and 1-peer exponential graphs, combining our BCD reduction with their large spectral gaps enables efficient decentralized learning even under heterogeneous communication delays.

## Impact Statement

We present a node assignment optimization method for decentralized learning under heterogeneous communication delays, which can be applied for training large-scale models (e.g., LLMs) over geographically distributed computing resources. A potential risk is that improving the efficiency of decentralized learning may lower the cost of large-scale decentralized learning, which could contribute to increased computational resource usage.

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

# A. Limitations and future work

First, limitations in our work are summarized.

**Static and symmetric communication delays.**    BTSP-MSR assumes that the communication delay function $d$ is given as a symmetric and time-invariant function. However, $d$ may be asymmetric or time-varying in practice. To address this concern, Appendix F.6 empirically shows that BTSP-MSR remains effective even under such communication-delay perturbations, but designing BCD minimization methods that explicitly handle such delays remains future work.

**Applicability of Theorem 3.**    Our tighter approximation ratio in Theorem 3 relies on structural assumptions on the Euclidean distance–based communication delay function (Assumption 1). In (T1) of Sec. 5, our experimental setting is constructed so that Assumption 1 holds rigorously; however, it does not necessarily hold in (T2), which uses a realistic NW scenario. As detailed in Appendix F.5, one alternative is to embed the $n$ nodes into a high-dimensional Euclidean space so that communication delays are well approximated by Euclidean distances between the embedded nodes; the existence of such an embedding is guaranteed by Bourgain's theorem (Bourgain, 1985). We empirically show that, even under this approximation, the prediction of Theorem 3 remains valid. This empirical validation helps justify the practical applicability of Theorem 3 in realistic NW scenarios, while a more rigorous analysis remains an important direction for future work.

**Restriction to multiplicative permutations.**    We use multiplicative permutations for the MSR step because they are efficiently enumerable in $O(n)$ and applicable to arbitrary node sizes $n$ (see Lemma 3). However, non-multiplicative permutations may also preserve the circulant structure when $n$ is not square-free (Muzychuk, 1995). Although we empirically investigated that no such permutations exist for exponential, 1-peer exponential, and sparse exponential graphs using (T1) and (T2), exploring broader classes of structure-preserving permutations remains future work.

Other remaining future work includes the following directions.

**Possibility of joint optimization of NW topology and node assignment.**    In this paper, we solve the node-assignment optimization problem for an NW topology represented as a circulant digraph. We challenge this issue because the estimated one-round runtime largely depends on the spectral gap of the NW topology; see Appendix B.4. Since communication-efficient NW topologies, such as exponential and 1-peer exponential graphs, are circulant digraphs, we construct a BCD minimization algorithm for a given circulant digraph. A promising future direction is the joint optimization of the NW topology and node assignment, with the goal of increasing the spectral gap while reducing BCD. However, this direction is substantially challenging because it leads to a nonlinear combinatorial optimization problem; see Appendix B.4.

**Broader synchronization costs.**    We consider only communication delay as the synchronization cost, while practical synchronization costs may also depend on bandwidth, congestion, computation speed, and other system-level factors. Extending the communication delay function $d$ to incorporate such costs is an important direction for practical deployment.

# B. Additional Details on Problem Formulation (Sec. 2)

## B.1. Influence of BCD on Decentralized Learning

As mentioned in Section 2, decentralized learning updates model parameters across $n$ nodes synchronously by iteratively performing (i) local updates within each node $i \in \{0, \cdots, n-1\}$ and (ii) partial mixing among connected nodes under a given NW topology. In real-world decentralized learning scenarios, computational nodes are often geographically distributed, leading to heterogeneous communication delays due to differences in physical distances and bandwidths. In decentralized learning under heterogeneous communication delays, (ii) the partial mixing is bottlenecked by the BCD, as shown in Figure 9.

Research on decentralized learning has mainly focused on NW topology design (Ying et al., 2021; Song et al., 2022; Takezawa et al., 2023; Niwa et al., 2025) and optimization algorithms (Zhang & Heusdens, 2017; Lin et al., 2021; Gao & Huang, 2020; Kong et al., 2021). However, BCD varies depending on the assignment of NW topology to physical NW. This is why we aim to optimize node assignment to minimize BCD. By performing node assignment optimization before the decentralized learning, the communication delay in each communication round can be minimized, thereby enabling communication-efficient decentralized learning throughout the entire learning process.

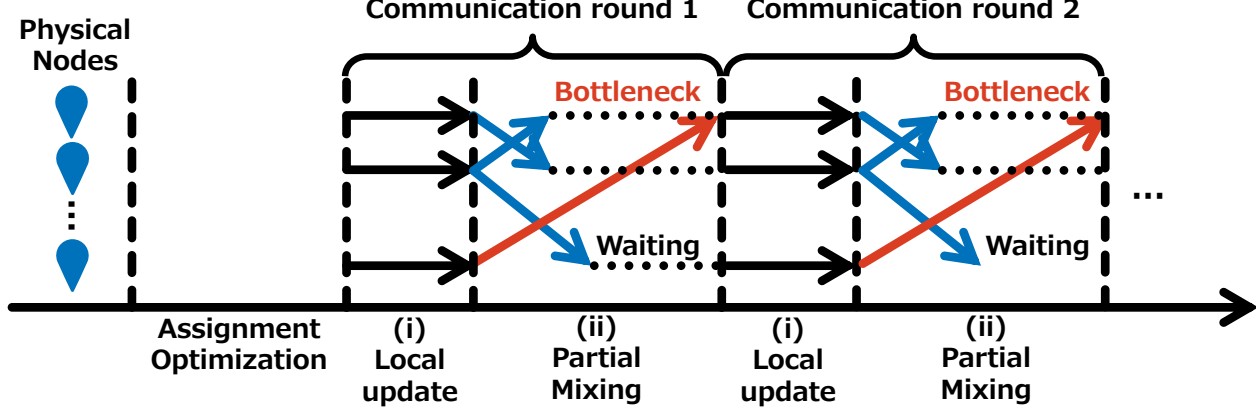

*Figure 9.* Procedure of decentralized learning. As an initialization step, node assignment optimization is performed. Given an NW topology (e.g., ring or exponential graph), it is assigned to the physical NW. Training then begins. When employing DSGD in (1), each communication round alternates between (i) local updates and (ii) partial mixing of local models. In the partial mixing, all nodes synchronously exchange information over the physical network. Due to heterogeneous communication delays between physical nodes, each round is constrained by the Bottleneck Communication Delay (BCD)—the longest delay among all required links—illustrated by the red arrow in the figure.

## B.2. Circulant Digraph Examples

Several circulant digraph examples used throughout this paper are detailed as follows.

**Ring Graph.** The ring graph is a static NW topology ($\tau = 1$) where each node is connected with its immediate neighbors. The mixing matrix is given by

$$W_{ij}^{(0)} = \begin{cases} 1/3 & ([j-i] \in \{0, 1, n-1\}) \\ 0 & (\text{otherwise}) \end{cases},$$

where the corresponding edge generator is given by $S^{(0)} = \{1, n-1\}$. Since we assume that the communication delay $d$ satisfies symmetry, BCD on the ring graph is equivalent to that on the directed ring graph generated from $S^{(0)} = 1$.

**Exponential Graph.** The exponential graph is a static NW topology ($\tau = 1$) where the skip-connection intervals between nodes increase exponentially. The mixing matrix is given by

$$W_{ij}^{(0)} = \begin{cases} \frac{1}{\lceil \log_2 n \rceil + 1} & ([j-i] \in \{0, 1, 2, \cdots, 2^{\lceil \log_2 n \rceil - 1}\}) \\ 0 & (\text{otherwise}) \end{cases},$$

*Table 1.* Specification comparison among ring, exponential, and 1-peer exponential graphs. The maximum degree indicates the maximum number of simultaneous connecting nodes per node. For 1-peer exponential graph, the spectral gap for any number of $n$ is given by (Niwa et al., 2025), which are interpreted in the expectation-form sense of Assumption 3.

| | (n1) Ring | (n2) Exp. | (n3) 1-peer Exp. | |
|---|---|---|---|---|
| Maximum Degree | 2 | $\lceil \log_2 n \rceil$ | 1 | |
| Spectral Gap $\lambda$ | $O\left(\frac{1}{n^2}\right)$ | $O\left(\frac{1}{\log_2 n}\right)$ | $\begin{cases} 1 \\ 1 - \max\limits_{i \in \{1, \ldots, n-1\}} \left\lvert \frac{1}{2^{\lceil \log_2(n) \rceil}} \frac{\sin(\pi i 2^{\lceil \log_2(n) \rceil}/n)}{\sin(\pi i/n)} \right\rvert \end{cases}$ | $\begin{matrix} n \text{ is a power of 2} \\ \text{otherwise} \end{matrix}$ |
| # of Nodes $n$ | arbitrary | arbitrary | A power of 2 | |
| # of Mixing Matrices $\tau$ | 1 | 1 | $\lceil \log_2 n \rceil$ | |

where the corresponding edge generator is given by $S^{(0)} = \{2^l \mid l = 1, 2, \cdots, 2^{\lceil \log_2 n \rceil - 1}\}$.

**1-peer Exponential Graph.** The 1-peer exponential graph is a dynamic NW topology ($\tau = \lceil \log_2 n \rceil$) which is a sequentialized variant of the exponential graph. The mixing matrix is given by

$$W_{ij}^{(l)} = \begin{cases} 1/2 & ([j-i] \in \{0, 2^l\}) \\ 0 & \text{(otherwise)} \end{cases},$$

$$(l \in \{\{0, 1, \cdots, \lceil \log_2 n \rceil - 1\}\}),$$

where the corresponding edge generator is given by $S^{(l)} = \{2^l\}$ for $l = 0, 1, \cdots, \lceil \log_2 n \rceil - 1$.

**Sparse Exponential Graph.** To investigate a relation between the sparsity of the NW topology and the performance of MSR (the second stage in BTSP-MSR), we introduce a sparse exponential graph defined as follows. The sparse exponential graph is a static NW topology ($\tau = 1$), where only the shortest and the longest connections in the exponential graph remain. From the symmetry of the communication delay $d$, the maximum value $2^{\lceil \log_2 n \rceil - 1} \in S^{(0)}$ is replaced with $n - 2^{\lceil \log_2 n \rceil - 1} \leq n/2$ in BTSP-MSR (Section 4(d)). Therefore, we use $2^{\lceil \log_2 n \rceil - 2}$ as the longest connection, and the mixing matrix is given by

$$W_{ij}^{(0)} = \begin{cases} 1/3 & ([j-i] \in \{0, 1, 2^{\lceil \log_2 n \rceil - 2}\}) \\ 0 & \text{(otherwise)} \end{cases},$$

and the corresponding edge generator is given by $S^{(0)} = \{1, 2^{\lceil \log_2 n \rceil - 2}\}$.

For the ring, exponential, and 1-peer exponential graphs, the expected spectral gap has been theoretically analyzed (Ying et al., 2021). Table 1 summarizes these results, where the maximum degree refers to the maximum number of edges incident to each node in the NW topologies; due to its simplicity, the spectral gap of the ring graph decreases rapidly with increasing $n$, whereas that of the exponential graph decreases more slowly, owing to its stronger connectivity. The 1-peer exponential graph, a sequentialized variant of the exponential graph, achieves the maximum expected spectral gap $\lambda = 1$ when $n$ is a power of two.

**B.3. Convergence Rate of DSGD**

The convergence rate of DSGD in (1) is given. In (Koloskova et al., 2021), the following assumptions are used for this aim.

**Assumption 3** (Spectral Gap in Expectation Form). There exists $\lambda \in (0, 1]$ such that

$$\mathbb{E}[\|XW - \bar{X}\|_F^2] \leq (1 - \lambda)\|X - \bar{X}\|_F^2$$

for any $X \in \mathbb{R}^{m \times n}$, where $\|\cdot\|_F$ denotes the Frobenius norm, $W = W^{(0)} \cdots W^{(\tau-1)}$ and averaged model parameters $\bar{X} = \frac{1}{n} X 1_n 1_n^\top$.

**Assumption 4** (Smooth Gradient). There exists $L > 0$ such that $\|\nabla f_i(\boldsymbol{x}) - \nabla f_i(\boldsymbol{y})\| \leq L\|\boldsymbol{x} - \boldsymbol{y}\|, \ \forall \boldsymbol{x}, \boldsymbol{y} \in \mathbb{R}^m$.

**Assumption 5** (Bounded Stochastic Noise). There exists $\sigma > 0$ such that $\|\nabla f_i(\boldsymbol{x}) - \nabla F_i(\boldsymbol{x})\| \leq \sigma^2, \ \forall \boldsymbol{x} \in \mathbb{R}^m$, where $\nabla f_i$ and $\nabla F_i$ denote full gradient and stochastic gradient at node $i$, respectively.

**Assumption 6** (Bounded Data Heterogeneity). There exists $\zeta > 0$ such that

$$\frac{1}{n} \sum_{i=0}^{n-1} \|\nabla f_i(\boldsymbol{x}) - \nabla f(\boldsymbol{x})\| \leq \zeta^2, \ \forall \boldsymbol{x} \in \mathbb{R}^m$$

where $f(\boldsymbol{x}) = \frac{1}{n} \sum_{i=0}^{n-1} f_i(\boldsymbol{x})$.

Assumption 3 is a unique assumption since it allows the spectral gap of dynamical NW topologies to be expressed in expectation form. The other assumptions are standard in the field of decentralized learning. The convergence rate of DSGD in (1) is given as follows:

**Theorem 4** (Convergence Rate of DSGD (Koloskova et al., 2021)). *Suppose that Assumptions 3-6 hold. For any $\epsilon > 0$, there exists a learning rate such that the accuracy $\frac{1}{R+1} \sum_{l=0}^{R} \mathbb{E}[\|\nabla f(\bar{\boldsymbol{x}}^{(l)})\|] \leq \epsilon$ can be achieved after*

$$R = O\left( \frac{\sigma^2}{n\epsilon^2} + \frac{\zeta\tau + \sigma\sqrt{\lambda\tau}}{\lambda\epsilon^{3/2}} + \frac{\tau}{\lambda\epsilon} \right) \cdot L f_0$$

*communication rounds where $\bar{\boldsymbol{x}}^{(l)} = \frac{1}{n} \sum_{i=0}^{n-1} \boldsymbol{x}_i^{(l)}$ denotes the global model parameter (average of local parameters $\boldsymbol{x}_0^{(l)}, \cdots, \boldsymbol{x}_{n-1}^{(l)}$), $f_0 = f(\bar{\boldsymbol{x}}^{(0)}) - f^*$ and $f^*$ denotes the minimum value of $f$.*

Theorem 4 shows that the convergence rate of DSGD improves with an increase in the spectral gap (in expectation form) of the mixing matrices. In Table 1, spectral gaps for each graph is summarized, as specified by $\lambda_{\text{ring}} = O\left(n^{-2}\right), \lambda_{\text{exp}} = O\left((\log_2 n)^{-1}\right)$, and

$$\lambda_{\text{1-p.exp.}} = \begin{cases} 1 & n \text{ is a power of 2} \\ 1 - \max_{i \in \{1, \ldots, n-1\}} \left| \frac{1}{2^{\lceil \log_2(n) \rceil}} \frac{\sin\left(\frac{\pi i 2^{\lceil \log_2(n) \rceil}}{n}\right)}{\sin\left(\frac{\pi i}{n}\right)} \right| & \text{otherwise} \end{cases}.$$

Substituting them into Theorem 4 results in

$$R_{\text{ring}} = O\left( \frac{\sigma^2}{n\epsilon^2} + \frac{\zeta n^2 + \sigma n}{\epsilon^{3/2}} + \frac{n^2}{\epsilon} \right) \cdot L f_0 \ (\forall n),$$

$$R_{\text{exp}} = O\left( \frac{\sigma^2}{n\epsilon^2} + \frac{\zeta \log_2 n + \sigma \sqrt{\log_2 n}}{\epsilon^{3/2}} + \frac{\log_2 n}{\epsilon} \right) \cdot L f_0 \ (\forall n),$$

$$R_{\text{1-p.exp.}} = O\left( \frac{\sigma^2}{n\epsilon^2} + \frac{\zeta \lceil \log_2 n \rceil + \sigma \sqrt{\lambda_{\text{1-p.exp.}} \lceil \log_2 n \rceil}}{\lambda_{\text{1-p.exp.}} \epsilon^{3/2}} + \frac{\lceil \log_2 n \rceil}{\lambda_{\text{1-p.exp.}} \epsilon} \right) \cdot L f_0$$

This observation underscores that employing a NW topology with a large spectral gap $\lambda$ improves convergence in decentralized learning (e.g., exponential and 1-peer exponential graphs). However, such NW topologies tend to have higher degrees, which can lead to large BCDs if node assignment is not optimized. This highlights the importance of our study, focusing on optimizing node assignment to reduce BCD under a circulant digraph, including exponential and 1-peer exponential graphs.

### B.4. Runtime Estimation to Explain the Importance of Using NW Topologies with Large Spectral Gaps

Based on Theorem 4, we can analytically estimate the time required to reach a target accuracy level. In this subsection, we clarify why we focus on BCD minimization under a fixed NW topology and the difficulty of the joint optimization of the NW topology and the node assignment.

Specifically, Theorem 4 provides an upper bound on the number of communication rounds required to reach a target accuracy level $\epsilon$, in terms of the spectral gap $\lambda$, the number of mixing matrices $\tau$, the data heterogeneity $\zeta$, variance of the stochastic gradient noise $\sigma^2$, Lipschitz smoothness $L$, number of nodes $n$, and initial optimality gap $f_0$. Then, the analytically estimated time required to reach a target level is obtained by

$$T_{\text{est}} = T_{\text{BCD}} \cdot R = O\left( \left( \frac{\sigma^2}{n\epsilon^2} + \frac{\zeta\tau + \sigma\sqrt{\lambda\tau}}{\lambda\epsilon^{3/2}} + \frac{\tau}{\lambda\epsilon} \right) \cdot f_0 L T_{\text{BCD}} \right).$$

Since hyperparameters (e.g., $\zeta, \sigma^2, L, f_0, n$) depend on the problem setting and are not directly controllable, it is reasonable to simplify the discussion by fixing them (e.g., $n = 87, \sigma^2 = 1, \zeta = 1, f_0 = 1, L = 1, \epsilon = 10^{-2}$). Under this setting, $T_{\text{est}}$ can be expressed as a

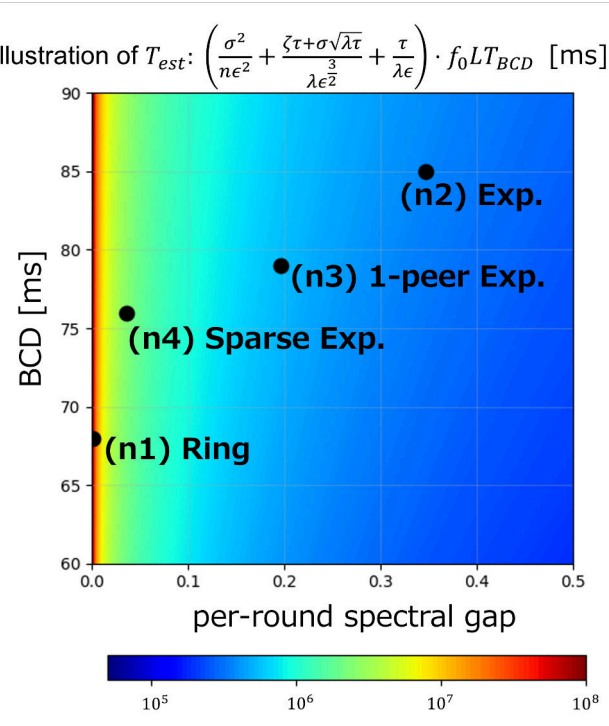

*Figure 10.* Visualization of analytically estimated runtime $T_{\text{est}}$ as a function of the per-round spectral gap $\lambda/\tau$ and BCD time $T_{\text{BCD}}$.

function of the per-round spectral gap $\lambda/\tau$ and the BCD time $T_{\text{BCD}}$, illustrated in Figure 10. In this figure, the total time can be reduced primarily by improving the spectral gap of the NW topology and secondarily by reducing the BCD. Motivated by this observation, we adopt the strategy of minimizing the BCD with fixed advanced NW topologies with large spectral gaps, such as exponential and 1-peer exponential graphs.

One could consider jointly optimizing the NW topology and the node assignment to directly minimize the analytically estimated runtime $T_{\text{est}}$; however, this joint optimization leads to a substantially more challenging problem. This is because the estimated runtime $T_{\text{est}}$ is a nonlinear function of the spectral gap $\lambda$, even when the other hyperparameters are fixed (e.g., $\sigma^2$, $L$, and $\zeta$). Moreover, designing NW topologies to increase the expected spectral gap $\lambda$ and reducing BCD through node assignment to decrease $T_{\text{BCD}}$ are both combinatorial optimization problems. Therefore, the joint optimization is a nonlinear combinatorial optimization problem, which is substantially more complex than the setting considered in this paper. We view the joint optimization as an important direction for future work, as mentioned in Appendix A.

## C. Proofs of Lemmas 1-3 and Theorem 1 (Sec. 4.1)

The proofs of Lemmas 1-3 and Theorem 1 in Sec. 4.1 are presented below.

### C.1. Proof of Lemma 1

*Proof.* We rewrite the edge set $E^{(l)}$ by the edge generator $S^{(l)}$ to prove this lemma. By the definition of the edge generator $S^{(l)}$, the edge set $E^{(l)}$ can be represented as $E^{(l)} = \{(i,j) \in V^2 \mid [j-i] \in S^{(l)}\}$. For an arbitrary edge $(i,j) \in E^{(l)}$, $s = [j-i]$ is an element of $S^{(l)}$. Since $[j-i]$ is equal to $j-i$ (when $i \leq j$) or $n+j-i$ (when $i > j$),

$$j = \begin{cases} i+s & (i \leq j) \\ i+s-n & (i > j) \end{cases},$$

then we have $j = [i+s]$. Hence, the edge set $E^{(l)}$ can be rewritten as

$$E^{(l)} = \{(i, [i+s]) \mid i \in V, s \in S^{(l)}\}. \tag{7}$$

Using this representation, the BCD objective $\varphi(\pi; \mathcal{E})$ can be rewritten as

$$\varphi(\pi; \mathcal{E}) = \frac{1}{\tau} \sum_{l=0}^{\tau-1} \max_{(i,j) \in E^{(l)}} d(\pi(i), \pi(j)) = \frac{1}{\tau} \sum_{l=0}^{\tau-1} \max_{i \in V, s \in S^{(l)}} d(\pi(i), \pi([i+s])).$$

$\square$

### C.2. Proof of Lemma 2

*Proof.* We use (3) and the triangle inequality of $d$ to prove this lemma. As mentioned before Lemma 2 in the main paper, $d$ returns the shortest communication delay between two physical nodes, and is not slower than communication through an intermediate physical node; thus, we can use the following triangle inequality

$$d(\pi(i), \pi([i+s])) \leq d(\pi(i), \pi([i+1])) + d(\pi([i+1]), \pi([i+s])).$$

By recurrently repeating this, we have

$$\begin{aligned} d(\pi(i), \pi([i+s])) &\leq \sum_{k=0}^{s-1} d(\pi([i+k]), \pi([i+k+1])) \\ &\leq s \cdot \max_{0 \leq k \leq s-1} d(\pi([i+k]), \pi([i+k+1])) \\ &\leq s \cdot \max_{j \in V} d(\pi(j), \pi([j+1])) \\ &= sB(\pi). \end{aligned}$$

Using this inequality, we have

$$\begin{aligned} \varphi(\pi; \mathcal{E}) &= \frac{1}{\tau} \sum_{l=0}^{\tau-1} \max_{i \in V, s \in S^{(l)}} d(\pi(i), \pi([i+s])) \\ &\leq \frac{1}{\tau} \sum_{l=0}^{\tau-1} \max_{i \in V, s \in S^{(l)}} sB(\pi) \\ &= B(\pi) \cdot \frac{1}{\tau} \sum_{l=0}^{\tau-1} \max_{s \in S^{(l)}} s \\ &= B(\pi) \cdot \underbrace{\frac{1}{\tau} \sum_{l=0}^{\tau-1} \max S^{(l)}}_{g(\mathcal{S})}. \end{aligned}$$

The term in the RHS $g(\mathcal{S})$ quantifies the worst-case trade-off, which we call the maximum skip-sum. As a simple example, when $\tau = 1$ and $S^{(0)} = \{1, s\}$ ($1 < s < n/2$), the maximum skip-sum is $g(\mathcal{S}) = s$. This means that minimizing only the BCD for the ring graph (skip width is then 1), $B(\pi) = \max_{i \in V} d(\pi(i), \pi([i+1]))$, inevitably leads to degradation for another skip width $s$, $\max_{i \in V} d(\pi(i), \pi([i+s]))$; in the worst case, the overall BCD $\varphi(\pi, (S^{(0)}))$ can be up to $g(\mathcal{S}) = s$ times larger than $B(\pi)$. $\square$

## C.3. Proof of Theorem 1

*Proof.* To prove (6), we proceed in the following two steps:

1. First, we prove that the permuted edge set $\sigma_c E = \{([ci], [cj]) \mid (i, j) \in E\}$ is generated from the permuted edge generator $\sigma_c S = \{[cs] \mid s \in S\}$.

2. Second, we prove that $\varphi(\pi \circ \sigma_c; \mathcal{E}) = \varphi(\pi; \sigma_c \mathcal{E})$ where $\sigma_c \mathcal{E} = \left(\sigma_c E^{(0)}, \cdots, \sigma_c E^{(\tau-1)}\right)$.

After proceeding these steps, we have $\varphi(\pi \circ \sigma_c; \mathcal{E}) = \varphi(\pi; \sigma_c \mathcal{E}) \leq B(\pi)g(\sigma_c \mathcal{S})$ by substituting $\mathcal{E} \leftarrow \sigma_c \mathcal{E}$ in (5): $\varphi(\pi; \mathcal{E}) \leq B(\pi)g(\mathcal{S})$.

**Step 1.** We begin by recalling that, for any edge $(i, j) \in E$, the target node $j$ can be written in the form $j = [i + s]$ for some $s \in S$, as shown in (7). Using this characterization, we can rewrite $\sigma_c E$ as follows:

$$\begin{aligned}
\sigma_c E &= \{([ci], [cj]) \mid (i, j) \in E\} \\
&= \{([ci], [c(i + s)]) \mid i \in V, s \in S\} \\
&= \{([ci], [[ci] + [cs]]) \mid i \in V, s \in S\}.
\end{aligned}$$

Since $\sigma_c : i \mapsto [ci]$ is a permutation (i.e., a bijection on $V$), the set $\{[ci] \mid i \in V\}$ is equal to $V$. Therefore, renaming $[ci]$ by $i'$, we have:

$$\begin{aligned}
\sigma_c E &= \{(i', [i' + [cs]]) \mid i' \in V, s \in S\} \\
&= \{(i', [i' + s']) \mid i' \in V, s' \in \sigma_c S\},
\end{aligned}$$

which shows that $\sigma_c E$ is generated from $\sigma_c S$ as claimed.

**Step 2.** We write out the LHS explicitly using the definition of $\varphi$:

$$\varphi(\pi \circ \sigma_c; \mathcal{E}) = \frac{1}{\tau} \sum_{l=0}^{\tau-1} \max_{i \in V, s \in S^{(l)}} d\Big(\pi([ci]), \pi([c(i + s)])\Big).$$

Following the same technique as in Step 1, we can rewrite this formula by using $i' = [ci]$ and $s' = [cs]$ $(\in \sigma_c S^{(l)})$:

$$\begin{aligned}
\varphi(\pi \circ \sigma_c; \mathcal{E}) &= \frac{1}{\tau} \sum_{l=0}^{\tau-1} \max_{i \in V, s \in S^{(l)}} d\Big(\pi([ci]), \pi([c(i + s)])\Big) \\
&= \frac{1}{\tau} \sum_{l=0}^{\tau-1} \max_{i' \in V, s' \in \sigma_c S^{(l)}} d\Big(\pi(i'), \pi([i' + s'])\Big),
\end{aligned}$$

which is equivalent to the RHS $\varphi(\pi; \sigma_c \mathcal{E})$ because $\sigma_c E^{(l)}$ is generated by $\sigma_c S^{(l)}$ for each $l$, as shown in Step 1. $\qquad\square$

## C.4. Proof of Lemma 3

Furthermore, since the communication delay function $d$ is defined in terms of a distance metric, we can leverage its symmetry property to further enhance MSR in Lemma 3.

**Lemma 3** (Symmetry of skip widths). *Without loss of generality, each edge generator $S^{(l)}$ can be restricted to $S^{(l)} \subset \{1, \ldots, \lfloor n/2 \rfloor\}$. In particular, the MSR term satisfies*

$$g(\mathcal{S}) = \frac{1}{\tau} \sum_{l=0}^{\tau-1} \max S^{(l)} \ \leq \ \lfloor n/2 \rfloor.$$

*Proof.* The objective in (3) remains invariant when each skip width $s$ by $n-s$, because

$$\begin{aligned}
\max_{i \in V} d(\pi(i), \pi([i + s])) &= \max_{i \in V} d(\pi([i + s]), \pi(i)) \\
&= \max_{i \in V} d(\pi(i), \pi([i + n - s])).
\end{aligned}$$

This implies that we can restrict the skip-width to the range $S^{(l)} \subset \{1, \cdots, \lfloor n/2 \rfloor\}$, and $g(\mathcal{S})$ can accordingly be bounded by $\lfloor n/2 \rfloor$. $\quad\square$

# D. Approximation Ratio Analyses of BTSP-MSR (Sec. 4.3)

In Appendix D.1, the proof of Theorem 2 is given. In Appendix D.2, the formal statements and proof of Theorem 3 and associated assumptions are presented. (Empirical validations associated with assumptions used in Theorem 3 are illustrated in Appendix E.5.) Since the upper bounds in Theorems 2 and 3 depend on the MSR term, Appendix D.4 further discusses the MSR term for the exponential and 1-peer exponential graphs.

## D.1. Proof of Theorem 2

*Proof.* Let $\pi_1$ be the 2-approximate solution of BTSP. For the optimal value of the BCD, we have $\min_\pi \varphi(\pi; \mathcal{E}) \leq \min_c \varphi(\pi_1 \circ \sigma_c; \mathcal{E})$. Using (6): $\varphi(\pi_1 \circ \sigma_c; \mathcal{E}) \leq B(\pi_1) g(\sigma_c \mathcal{S})$, we have

$$\min_\pi \varphi(\pi; \mathcal{E}) \; \leq \; \min_c \varphi(\pi_1 \circ \sigma_c; \mathcal{E}) \; \leq \; B(\pi_1) \min_c g(\sigma_c \mathcal{S}).$$

Since $\pi_1$ is 2-approximate solution of BTSP,

$$B(\pi_1) \; \leq \; 2\min_\pi B(\pi) \; \leq \; 2\min_\pi \varphi(\pi; \mathcal{E}).$$

and therefore,

$$\min_c \varphi(\pi_1 \circ \sigma_c; \mathcal{E}) \; \leq \; 2\min_c g(\sigma_c \mathcal{S}) \min_\pi \varphi(\pi; \mathcal{E}),$$

which shows that BTSP-MSR achieves a $2\min_c g(\sigma_c \mathcal{S})$-approximation ratio. $\qquad\square$

The assumption $\min_\pi B(\pi) \leq \min_\pi \varphi(\pi; \mathcal{E})$ holds in most NW topologies. For example, any static ($\tau = 1$) circulant digraphs that contain the ring graph as a subgraph satisfies this assumption. This is because the BCD $\varphi(\pi; \mathcal{E})$ includes all ring edges as a subset of its maximization, and therefore cannot be smaller than the optimal BTSP objective defined solely on the ring graph. For the 1-peer exponential graph, the assumption is satisfied when the number of nodes $n$ is odd. When $n$ is odd, every skip width $2^l$ is coprime to $n$, so each communication round induces a single directed ring graph; consequently, the communication delay in every communication round is not smaller than the BTSP objective, yielding the assumption $\min_\pi B(\pi) \leq \min_\pi \varphi(\pi; \mathcal{E})$.

## D.2. Formal Statement and Proof of Theorem 3

Firstly, formal statements of Assumptions used in Theorem 3 are given.

**Assumption 1** ((Formal) Euclidean Distance-based $d$)**.** The communication delay function $d$ is proportional to the Euclidean distance with a constant communication overhead:

$$d(i, j) = \kappa \|u_i - u_j\| + b, \qquad (\kappa > 0, \; b \geq 0 : \text{ const.}, \forall i, j)$$

**Assumption 2** ((Formal) Bounded Maximum Edge-Direction Correlation)**.** Let $u_0, \cdots, u_{n-1}$ denote the random positions of $n$ physical nodes. Let $\pi_1$ be a BTSP solution, and let $c \in \arg\min_c g(\sigma_c \mathcal{S})$. For each $i \in V$, define the normalized edge-direction vector

$$e_i = \frac{u_{\pi_1([i+1])} - u_{\pi_1(i)}}{\max_{j \in V} \left\| u_{\pi_1([j+1])} - u_{\pi_1(j)} \right\|}.$$

For any integer $s \geq 1$, define the maximum edge-direction correlation

$$h(s) = 2\max_{i \in V} \sum_{0 \leq j < k \leq s-1} \left\langle e_{[i+j]}, e_{[i+k]} \right\rangle.$$

Then, for any $\delta \in (0, 1)$, there exists a constant $\beta > 0$ such that, for every skip width: $s \in \bigcup_{l=0}^{\tau-1} \sigma_c S^{(l)}$,

$$\Pr(h(s) \leq \beta) \geq 1 - \delta.$$

The maximum edge-direction correlation $h(s)$ quantifies the strongest directional alignment among any $s$ consecutive communication links (between physical nodes) along the BTSP solution $\pi_1$. Since each inner product $\langle e_{[i+j]}, e_{[i+k]} \rangle$ measures the directional similarity between two links, a large value of $h(s)$ indicates that many links are nearly collinear, yielding an almost straight trajectory in the physical space. However, in practical NW settings, physical nodes are typically distributed in a randomized manner, making such sustained collinearity unlikely; consequently, $h(s)$ is expected to remain small. Motivated by this observation, Assumption 2 postulates that $h(s)$ is bounded by a constant $\beta$ with high probability.

**Theorem 3** ((Formal) Approximation Ratio of BTSP-MSR under Practical NW Scenarios). *Let $\pi_1$ be the output of a 2-approximation algorithm for BTSP, $\pi^\star = \pi_1 \circ \sigma_c$ be the output of BTSP-MSR, and $|\mathcal{S}| = \sum_l |S^{(l)}|$ be the number of skip widths. Suppose that Assumptions 1–2 hold. Then, for any $\delta > 0$, with probability at least $1 - |\mathcal{S}|\delta$,*

$$\varphi(\pi^\star; \mathcal{E}) \leq 2 \min_c \sqrt{g(\sigma_c \mathcal{S}) + \beta} \big( \min_\pi \varphi(\pi; \mathcal{E}) - b \big) + b.$$

*Proof.* By Assumption 1, we obtain

$$d(\pi^*(i), \pi^*([i+s])) = \kappa \big\| u_{\pi^*([i+s])} - u_{\pi^*(i)} \big\| + b.$$

Let $i' = [ci]$ and $s' = [cs]$. Then $\pi^*(i) = \pi_1(i')$ and $\pi^*([i+s]) = \pi_1([i'+s'])$. Hence, we have

$$u_{\pi^*([i+s])} - u_{\pi^*(i)} = \sum_{k=0}^{s'-1} \big( u_{\pi_1([i'+k+1])} - u_{\pi_1([i'+k])} \big).$$

Using Assumption 1, $B(\pi_1) = \max_{j \in V} d(\pi_1(j), \pi_1([j+1])) = \kappa \max_{j \in V} \| u_{\pi_1([j+1])} - u_{\pi_1(j)} \| + b$. Associated with this, we introduce $L = \max_{j \in V} \| u_{\pi_1([j+1])} - u_{\pi_1(j)} \| \left( = \frac{B(\pi_1) - b}{\kappa} \right)$. By the definition of the normalized edge vectors $e_i$, we have

$$u_{\pi_1([i'+k+1])} - u_{\pi_1([i'+k])} = L\, e_{[i'+k]}.$$

Therefore, we get

$$u_{\pi^*([i+s])} - u_{\pi^*(i)} = L \sum_{k=0}^{s'-1} e_{[i'+k]}.$$

Consequently, we have

$$\left\| \sum_{k=0}^{s'-1} e_{[i'+k]} \right\|^2 = \sum_{k=0}^{s'-1} \| e_{[i'+k]} \|^2 + 2 \sum_{0 \leq j < k \leq s'-1} \langle e_{[i'+j]}, e_{[i'+k]} \rangle.$$

Since $\| e_i \| \leq 1$ for all $i$, we have

$$\sum_{k=0}^{s'-1} \| e_{[i'+k]} \|^2 \leq s'.$$

Moreover, by Assumption 2, with probability at least $1 - \delta$, uniformly over $i'$, we have

$$2 \sum_{0 \leq j < k \leq s'-1} \langle e_{[i'+j]}, e_{[i'+k]} \rangle \leq \beta.$$

Combining the above inequalities, with probability at least $1 - \delta$, for all $i, s$, we have

$$\left\| \sum_{k=0}^{s'-1} e_{[i'+k]} \right\| \leq \sqrt{s' + \beta}.$$

Hence, we get

$$\big\| u_{\pi^*([i+s])} - u_{\pi^*(i)} \big\| \leq \frac{B(\pi_1) - b}{\kappa} \sqrt{s' + \beta}.$$

Multiplying by $\kappa(> 0)$ and adding $b(\geq 0)$, we obtain

$$d(\pi^*(i), \pi^*([i+s])) \leq (B(\pi_1) - b) \sqrt{s' + \beta} + b,$$

where we used Assumption 1.

Now fix a communication round $l \in \{0, \ldots, \tau - 1\}$. By a union bound over $s \in S^{(l)}$, with probability at least $1 - |S^{(l)}|\delta$, the above inequality holds simultaneously for all $i \in V$ and $s \in S^{(l)}$. Thus,

$$\max_{i \in V,\, s \in S^{(l)}} d(\pi^*(i), \pi^*([i+s])) \leq (B(\pi_1) - b) \sqrt{\max \sigma_c S^{(l)} + \beta} + b.$$

Recalling that

$$\varphi(\pi^*; \mathcal{E}) = \frac{1}{\tau} \sum_{l=0}^{\tau-1} \max_{i \in V, \, s \in S^{(l)}} d(\pi^*(i), \pi^*([i+s])),$$

and taking a union bound over $l$, with probability at least $1 - |\mathcal{S}|\delta$,

$$\varphi(\pi^*; \mathcal{E}) \leq \frac{B(\pi_1) - b}{\tau} \sum_{l=0}^{\tau-1} \sqrt{\max \sigma_c S^{(l)} + \beta} + b.$$

Since $x \mapsto \sqrt{x}$ is concave on $\mathbb{R}_{\geq 0}$, Jensen's inequality yields

$$\frac{1}{\tau} \sum_{l=0}^{\tau-1} \sqrt{\max \sigma_c S^{(l)} + \beta} \leq \sqrt{g(\sigma_c \mathcal{S}) + \beta}.$$

Therefore, we get

$$\varphi(\pi^*; \mathcal{E}) \leq (B(\pi_1) - b)\sqrt{g(\sigma_c \mathcal{S}) + \beta} + b.$$

Finally, since $\pi_1$ is a 2-approximation for BTSP,

$$B(\pi_1) - b \leq 2\big(\min_\pi \varphi(\pi; \mathcal{E}) - b\big),$$

which completes the proof. $\qquad\square$

## D.3. Interpretation of $h(s)$, $\beta$, and $\delta$

We further clarify the three quantities in Theorem 3. The quantity $h(s)$ is the maximum correlation in communication directions in any consecutive physical nodes along the BTSP tour. The parameter $\beta$ is a high-probability upper bound on $h(s)$, and $\delta$ is the probability of failure in the high-probability analysis. Among these quantities, $h(s)$ may be difficult to interpret. To provide intuition, we consider three scenarios of node positions.

**Scenario 1: Aligned in a straight line.** This corresponds to the most conservative scenario discussed in Theorem 2, where physical nodes are aligned in a straight line. In this case, $h(s)$ can be computed analytically as $\max_s h(s) = O(n^2)$, as shown below.

Consider $n$ physical nodes placed on a line whose positions are $x_i = i$ for $i = 0, 1, \ldots, n-1$. We first show that the optimal bottleneck value $B(\pi)$ of a BTSP solution is 2 in this example. Since there are only $n-1$ pairs of nodes at distance 1, a solution using only edges of length at most 1 cannot connect all $n$ nodes and return to the starting node. Therefore, the optimal value must be at least 2.

Conversely, a feasible solution with bottleneck value 2 can be explicitly constructed. If $n$ is even, define $\pi$ by

$$\pi(j) = \begin{cases} 2j, & 0 \leq j \leq \frac{n}{2} - 1, \\ 2(n-j) - 1, & \frac{n}{2} \leq j \leq n - 1. \end{cases}$$

If $n$ is odd, define $\pi$ by

$$\pi(j) = \begin{cases} 2j, & 0 \leq j \leq \frac{n-1}{2}, \\ 2(n-j) - 1, & \frac{n+1}{2} \leq j \leq n - 1. \end{cases}$$

For both definitions, consecutive nodes in each part of the ordering are at distance 2. The two boundary edges have length 1: when $n$ is even,

$$\left| \pi\left(\frac{n}{2}\right) - \pi\left(\frac{n}{2} - 1\right) \right| = 1, \qquad |\pi(0) - \pi(n-1)| = 1,$$

and when $n$ is odd,

$$\left| \pi\left(\frac{n+1}{2}\right) - \pi\left(\frac{n-1}{2}\right) \right| = 1, \qquad |\pi(0) - \pi(n-1)| = 1.$$

Thus, all consecutive edges induced by $\pi$ have length at most 2, including the edge returning to the starting node. Hence, there exists a BTSP solution with $B(\pi) = 2$. Combining this with the lower bound above, the optimal bottleneck value of the BTSP solution is 2.

We next evaluate $h(s)$ along this BTSP solution. Let $e_i$ be the normalized edge-direction vector defined in Assumption 2. In the first part of the above ordering, consecutive edges have length 2 and move in the same direction along the line. Since the maximum edge length

of this BTSP solution is 2, these normalized vectors have norm 1 and are identical. Hence, for any $s \leq \lfloor (n-1)/2 \rfloor$, we can choose a consecutive segment of length $s$ contained in this first part. For such a segment,

$$\sum_{0 \leq j < k \leq s-1} \langle e_{[i+j]}, e_{[i+k]} \rangle = \binom{s}{2}.$$

Therefore, by the definition of $h(s)$,

$$h(s) \geq 2 \binom{s}{2} = s(s-1).$$

Since $\|e_i\| \leq 1$ for all $i$, we also have

$$h(s) \leq 2 \binom{s}{2} = s(s-1).$$

Thus, for $s \leq \lfloor (n-1)/2 \rfloor$, we obtain

$$h(s) = \Theta(s^2).$$

Taking $s = \Theta(n)$ gives

$$\max_s h(s) = \Theta(n^2).$$

**Scenario 2: Placed on concentric circles.** This corresponds to a less conservative scenario where physical nodes are spread over a two-dimensional region. In this case, $h(s)$ can also be analytically bounded as $h(s) = O(n)$, as shown below.

Let $n = 4k^2$ physical nodes be placed evenly on $2k$ concentric circles. Specifically, for $0 \leq a, b \leq 2k - 1$, define

$$r_a = 1 + \frac{a}{2k}, \qquad \theta_b = \frac{2\pi b}{2k},$$

and place the physical node $u_{2ka+b}$ at

$$u_{2ka+b} = r_a(\cos \theta_b, \sin \theta_b).$$

All physical nodes are contained in a disk of radius 2, and hence the distance between any two physical nodes is $O(1)$.

We then construct a BTSP solution $\pi_1$ for these physical nodes by repeating the construction used in Scenario 1 along each radial line. For a fixed angular index $b$, the physical nodes

$$u_{2k \cdot 0 + b}, \ u_{2k \cdot 1 + b}, \ \ldots, \ u_{2k(2k-1)+b}$$

are aligned on the same straight line. Starting from $b = 0$, we apply the one-skip-and-return ordering from Scenario 1 to this radial line:

$$u_{2k \cdot 0 + b}, \ u_{2k \cdot 2 + b}, \ \ldots, \ u_{2k(2k-2)+b}, \ u_{2k(2k-1)+b}, \ u_{2k(2k-3)+b}, \ \ldots, \ u_{2k \cdot 1 + b}.$$

After reaching $u_{2k \cdot 1 + b}$, the solution moves to the adjacent radial line by using the edge

$$u_{2k \cdot 1 + b} \rightarrow u_{2k \cdot 1 + (b+1)}.$$

On the next radial line, we apply the same construction in the opposite direction, and repeat this procedure for $b = 0, 1, \ldots, 2k - 1$. This yields a BTSP solution that visits all physical nodes and returns to the starting node.

We next show the optimality of this BTSP solution $\pi_1$. A BTSP solution visiting all physical nodes cannot change the angular index only on the innermost circle; otherwise, the physical nodes on the innermost circle would be disconnected from the remaining physical nodes in the BTSP solution. Therefore, any BTSP solution must contain at least one edge connecting different angular indices outside the innermost circle. Since the length of such an edge increases with the radius, the shortest possible edge of this type is attained on the second innermost circle, whose length is

$$2r_1 \sin \frac{\pi}{2k} = M_0.$$

Hence, for any BTSP solution $\pi$, we have

$$B(\pi) \geq M_0.$$

Since the constructed BTSP solution $\pi_1$ achieves this lower bound, $\pi_1$ is the optimal BTSP solution.

We next evaluate $h(s)$ along this BTSP solution $\pi_1$. Let $e_i$ be the normalized edge-direction vector defined in Assumption 2. By the definition of $e_i$, for any $i \in \{0, 1, \cdots, n - 1\}$ and $s \geq 1$, we have

$$\sum_{t=0}^{s-1} e_{[i+t]} = \frac{u_{\pi_1([i+s])} - u_{\pi_1(i)}}{M_0}.$$

Since all physical nodes are contained in a disk of radius 2, the numerator is $O(1)$. Moreover,

$$M_0 = \Theta\left(\frac{1}{k}\right).$$

Hence,

$$\max_{i \in V} \left\| \sum_{t=0}^{s-1} e_{[i+t]} \right\| = O(k),$$

and therefore

$$\max_{i \in V} \left\| \sum_{t=0}^{s-1} e_{[i+t]} \right\|^2 = O(k^2) = O(n).$$

Finally, by expanding the squared norm,

$$\left\| \sum_{t=0}^{s-1} e_{[i+t]} \right\|^2 = \sum_{t=0}^{s-1} \|e_{[i+t]}\|^2 + 2 \sum_{0 \leq j < \ell \leq s-1} \langle e_{[i+j]}, e_{[i+\ell]} \rangle.$$

Since the first term on the right-hand side is nonnegative, we obtain

$$h(s) \leq \max_{i \in V} \left\| \sum_{t=0}^{s-1} e_{[i+t]} \right\|^2 = O(n).$$

Thus, in this two-dimensional placement, $h(s)$ grows at most linearly in $n$, which is substantially smaller than the $\Theta(n^2)$ scaling in Scenario 1.

**Scenario 3: Uniform random distribution.** We consider this setting close to practical NW scenarios. Although $h(s)$ is difficult to evaluate analytically, it can be estimated empirically from randomly generated positions, yielding a high-probability bound, as in Appendix E.5. Using this estimate, Appendix E.5 shows that the practical upper bound in Theorem 3 is significantly tighter than that of Theorem 2.

The above discussion also explains why we introduce randomness in Theorem 3 and Assumption 2. For given node positions and communication delays, one can derive an instance-dependent upper bound using the observed maximum value of $h(s)$. For example, for the Ebone NW, we estimate a tighter upper bound in Appendix F.5. However, we introduce randomness to avoid rare but adversarial node positions that would otherwise dominate the bound. As shown in Appendix E.5, small $\delta$ leads to large estimates of $\beta$, reflecting these adversarial events. The high-probability formulation in Theorem 3 effectively excludes such statistically unlikely patterns, resulting in a much tighter bound.

### D.4. Analytical Estimation of the MSR Term for Exponential and 1-Peer Exponential Graphs

The approximation ratios in Theorems 2 and 3 both include the MSR term $\min_c g(\sigma_c \mathcal{S})$. Lemma 3 gives a general worst-case bound $\min_c g(\sigma_c \mathcal{S}) \leq \lfloor n/2 \rfloor = O(n)$ for arbitrary circulant digraphs, however, for specific NW topologies, this term may have a smaller asymptotic order. In this subsection, we further analytically investigate the MSR term for the exponential and 1-peer exponential graphs.

First, recall that the MSR term is defined as

$$g(\mathcal{S}) = \frac{1}{\tau} \sum_{l=0}^{\tau-1} \max S^{(l)}.$$

With the multiplicative permutation $\sigma_c$, it can be rewritten as

$$g(\sigma_c \mathcal{S}) = \frac{1}{\tau} \sum_{l=0}^{\tau-1} \max_{s \in S^{(l)}} [cs],$$

where $[cs] = cs \bmod n$. Furthermore, using the symmetry of the communication delay function $d$, we can replace each skip width $[cs]$ by $\min\{[cs], n - [cs]\}$, as discussed in Lemma 3. Therefore, in the worst case, $\min_c g(\sigma_c \mathcal{S}) = O(n)$.

**Exponential graph.** For the exponential graph, the edge generator is given by

$$S^{(0)} = \{2^0, 2^1, \ldots, 2^{\lceil \log_2 n \rceil - 1}\}, \qquad \tau = 1.$$

Using the symmetry of the distance function $d$, the MSR term for the exponential graph is

$$g(\sigma_c \mathcal{S}) = \max_{s \in S^{(0)}} \min\{[cs], n - [cs]\}.$$

However, due to the presence of the modulo operation, deriving a simple closed-form expression for $\min_c g(\sigma_c \mathcal{S})$ is difficult. Instead, we show that $\min_c g(\sigma_c \mathcal{S}) = \Theta(n)$ for $n = 2^k - 1$, indicating that the asymptotic order of the MSR term is unchanged, while leaving open the possibility of improvement at the constant-factor level.

**Lemma 4.** *Consider the exponential graph with $n = 2^k - 1$ for an integer $k \geq 2$. Then, the following lower bound on the MSR term holds:*

$$\min_c g(\sigma_c \mathcal{S}) \geq \frac{n+1}{4}.$$

*Proof.* For the exponential graph with $n = 2^k - 1$, we have

$$S^{(0)} = \{2^0, 2^1, \ldots, 2^{k-1}\}.$$

For any $c \in \{1, \ldots, n-1\}$ coprime to $n$, the elements $[c2^l]$ correspond to cyclic shifts of the $k$-bit binary representation of $c$:

$$c = \sum_{r=0}^{k-1} b_r 2^r, \qquad b_r \in \{0, 1\}.$$

Since $n = 2^k - 1$, we have $2^k \equiv 1 \pmod{n}$. Thus,

$$[2c] = \left[\sum_{r=0}^{k-1} b_r 2^{r+1}\right] = b_{k-1} + \sum_{r=0}^{k-2} b_r 2^{r+1}.$$

This means that multiplication by 2 modulo $n$ cyclically shifts the $k$-bit representation of $c$ by one position. Repeating this operation, multiplication by $2^l$ modulo $n$ cyclically shifts the $k$-bit representation of $c$ by $l$ positions. Hence, $\sigma_c S^{(0)}$ consists of all cyclic shifts of $c$.

Since $0 < c < 2^k - 1$, the binary representation of $c$ is neither all-zero nor all-one. Thus, there exists a cyclic shift whose first two bits are in $\{01, 10\}$. Such a value is at least $2^{k-2}$ and at most $3 \cdot 2^{k-2} - 1$. Therefore, after applying the symmetry $[cs] \mapsto \min\{[cs], n - [cs]\}$, there exists a skip width whose value is at least $2^{k-2}$. Since $2^{k-2} = (n+1)/4$, we obtain

$$g(\sigma_c \mathcal{S}) = \max_{s \in S^{(0)}} \min\{[cs], n - [cs]\} \geq \frac{n+1}{4}.$$

Because this holds for any admissible $c$, we have

$$\min_c g(\sigma_c \mathcal{S}) \geq \frac{n+1}{4}.$$

$\square$

Lemma 4 shows that, for the exponential graph, $\min_c g(\sigma_c \mathcal{S}) = \Theta(n)$ in this setting. Therefore, although the constant factor may be improved by the multiplicative permutation, the asymptotic order of the MSR term remains linear in $n$.

**1-peer exponential graph.** For the 1-peer exponential graph, each communication round uses one skip width:

$$S^{(l)} = \{2^l\}, \qquad l = 0, 1, \ldots, \lceil \log_2 n \rceil - 1.$$

Thus, the MSR term is an average over $\tau = \lceil \log_2 n \rceil$ terms:

$$g(\sigma_c \mathcal{S}) = \frac{1}{\tau} \sum_{l=0}^{\tau-1} \min\{[c2^l], n - [c2^l]\}.$$

Each term is bounded by $O(n)$, and $\tau = O(\log n)$.

For the 1-peer exponential graph, by taking $c = 1$, we have

$$g(S) = \frac{1}{\tau} \sum_{l=0}^{\tau-1} \min\{2^l, n - 2^l\} \leq \frac{1}{\tau} \sum_{l=0}^{\tau-1} 2^l = O\left(\frac{n}{\log n}\right).$$

Therefore,

$$\min_c g(\sigma_c \mathcal{S}) \leq g(\mathcal{S}) = O\left(\frac{n}{\log n}\right).$$

This result indicates that the 1-peer exponential graph admits an MSR term with a smaller asymptotic order, than the general worst-case bound $O(n)$.

# E. Additional Experimental Setups and Results Regarding (T1) in Sec. 5

This appendix section provides additional experimental details and results corresponding to (T1) in Sec. 5. We first describe the solver configuration in Appendix E.1. We then report additional experiments comparing BTSP-MSR with simulated annealing in Appendix E.2. We also evaluate the suboptimality of BTSP-MSR on small-scale instances in Appendix E.3. Next, we empirically investigate the computational time and the impact of the NW topology of BTSP-MSR in Appendix E.4. Further, in Appendix E.5, we empirically validate the assumptions and practical upper bounds used in Theorem 3. Finally, we present qualitative visualizations of node assignments to illustrate how BTSP-MSR reduces bottleneck communication delays in Appendix E.6.

## E.1. SCIP Configuration

While SCIP allows configurable settings, we set the initial solution to the identity permutation (i.e., $\pi = $ id) and disabled the presolving. This choice was because, with presolving enabled, the solver occasionally consumed the entire 30-minute time budget during presolving, without returning any better solution than the initial solution.

## E.2. Comparison with Simulated Annealing

We conducted additional experiments in (T1) comparing BTSP-MSR with simulated annealing (Kirkpatrick et al., 1983), a heuristic that starts with broad exploration and gradually shifts toward local refinement. Since this paper particularly focuses on the effectiveness of BTSP-MSR in large-scale settings, we evaluated the methods for $n \in \{64, \ldots, 128\}$ under the simulated physical NWs used in (T1).

For simulated annealing, we used a random initial assignment and swap-based candidate generation. The transition from exploration to refinement is controlled by a temperature parameter: a candidate was accepted if it improved the BCD objective $\phi(\pi; \mathcal{E})$ in Eq. (2); otherwise, it was accepted with probability $\exp(-\Delta/T)$, where $\Delta$ is the increase in the BCD objective and $T$ is the current temperature. We used $T = 1.0$, the geometric cooling schedule $T \leftarrow 0.8T$, and 100 iterations.

Figure 11 shows the comparison results. From the results, we observe that BTSP-MSR outperforms simulated annealing in most cases, especially for larger numbers of nodes. This result suggests that BTSP-MSR remains effective in large-scale settings by exploiting the circulant structure of the target NW topology.

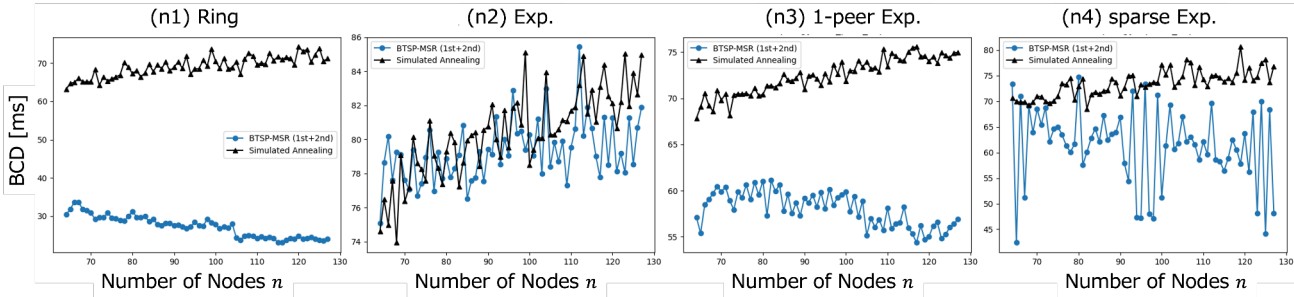

*Figure 11.* Comparison between BTSP-MSR and simulated annealing under the simulated physical NWs used in (T1), evaluated for $n \in \{64, \ldots, 128\}$. BTSP-MSR outperforms simulated annealing in most cases, especially for larger numbers of nodes.

## E.3. Empirical Suboptimality Evaluation Using Small-Scale Instances

In the experiments of (T1), SCIP becomes impractical for large-scale instances due to its exponential computational cost. However, for small-scale instances, SCIP can sometimes compute exact optimal solutions within the time limit. Such cases allow us to empirically evaluate the suboptimality of BTSP-MSR by directly comparing its achieved BCD with the optimal value computed by SCIP.

Specifically, we evaluate small-scale settings with $n \in [10, 15]$ under the same simulated physical NWs as in (T1). Figure 12 compares BTSP-MSR ($1^{\text{st}}+2^{\text{nd}}$), the exact optimal values computed by SCIP, and the probabilistic upper bound in Theorem 3 with $\delta = 0.05$.

The results show that BTSP-MSR achieves near-optimal BCDs for all tested NW topologies; the BCDs achieved by BTSP-MSR are much closer to the exact optimal values computed by SCIP than to the probabilistic upper bounds in Theorem 3. This suggests that the actual suboptimality of BTSP-MSR is small in these small-scale settings.

## E.4. Additional Empirical Evaluations of BTSP-MSR

**Computational Time Investigation.** Figure 13 illustrates the average running time of BTSP-MSR over five independent runs for four circulant digraphs with $n \in \{8, 9, \cdots, 2047\}$ nodes. The running time closely follows the analytical complexity $O(n^2 \log n)$. Even with $n = 2047$ nodes, BTSP-MSR completes within two minutes, demonstrating that BTSP-MSR runs sufficiently fast and is practical for large-scale decentralized learning.

**Effectiveness of BTSP-MSR for NW Topologies Employing Long Skip Connection.** To further characterize the NW topologies for which BTSP-MSR is effective, we conduct an additional experiment on circulant digraphs that contain a single long

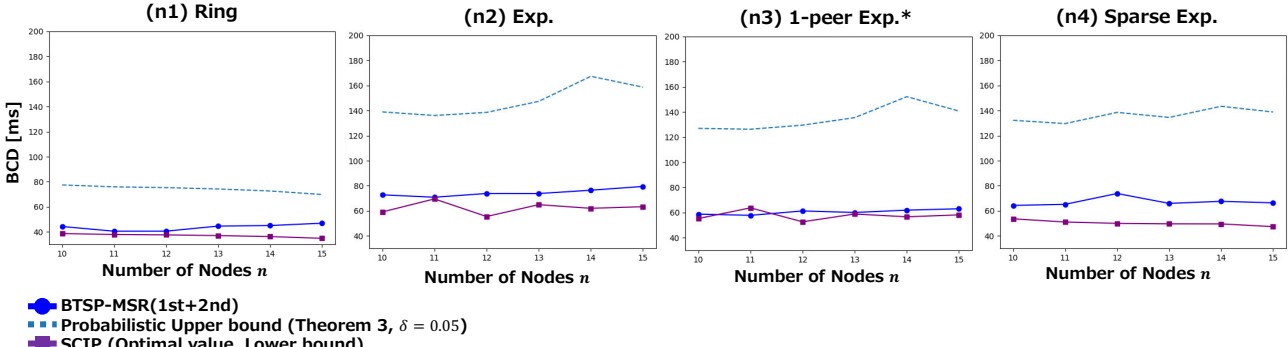

*Figure 12.* Suboptimality evaluation on small-scale instances with $n \in [10, 15]$. We report the BCD achieved by BTSP-MSR ($1^{st}+2^{nd}$), the exact optimal values computed by SCIP, and the probabilistic upper bound in Theorem 3 with $\delta = 0.05$. *Since SCIP cannot compute the exact optimal solution with $n \leq 11$, BTSP-MSR ($1^{st}+2^{nd}$) outperforms SCIP with $n = 11$. For other NW topologies, SCIP can compute the exact optimal solution.

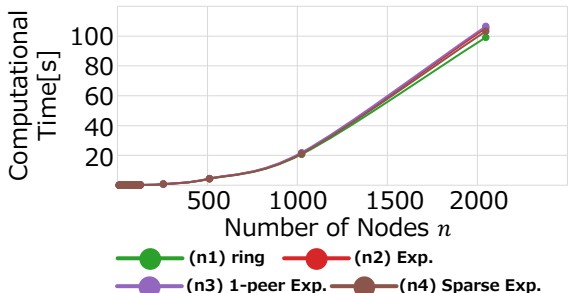

*Figure 13.* Running time of BTSP-MSR for four circulant digraphs, which closely follow the analytical complexity $O(n^2 \log n)$.

skip connection. In Sec. 4-5, we considered that long skip connections can make BTSP-MSR particularly beneficial, because MSR may substantially reduce the term $g(\sigma_c \mathcal{S})$. To validate this hypothesis, we consider a family of static NW topologies with $\tau = 1$ and $S^{(0)} = \{1, s\}$, and vary the additional skip width $s$ to examine how the resulting BCD changes as $s$ increases.

Figure 14 reports the achieved BCD $\varphi(\pi; \mathcal{E})$ for $n = 127$ nodes and $s \in \{3, \ldots, 63\}$ on the simulated geographically embedded NWs used in Sec. 5 (T1). Notably, even for large $s$, the BCD does not increase proportionally with the skip width and remains well controlled. This observation empirically supports our claim that BTSP-MSR mitigates the adverse effect of long skip connections through MSR minimization.

A particularly illustrative case is $S^{(0)} = \{1, (n-1)/2\}$ on $n$ nodes, where $(n-1)/2$ corresponds to a maximally long skip connection. In this setting, MSR can dramatically tighten the approximation ratio: choosing $c = 2$ yields $\sigma_2 S^{(0)} = \{2, n-1\}$. By symmetry of the communication delay, $\{2, n-1\}$ can be treated as $\{1, 2\}$, and hence the approximation ratio decreases from $2g(S) = 2(n-1)/2 = n-1$ to $2 \min_c g(\sigma_c S) = 4$. This example represents a topology in which MSR is maximally effective. The rightmost setting ($s = 63$) in Figure 14 corresponds to this case, where both the theoretical upper bound and the observed BCD decrease substantially.

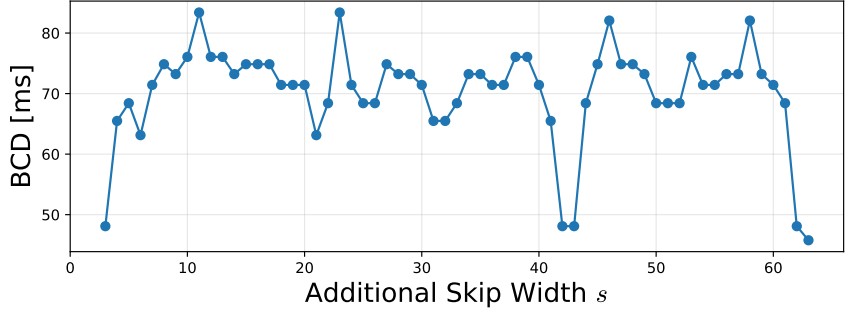

*Figure 14.* Achieved BCD $\varphi(\pi^*; \mathcal{E})$ for circulant digraphs with $\tau = 1$, $S^{(0)} = \{1, s\}$, evaluated for $n = 127$ and $s \in \{3, \ldots, 63\}$.

### E.5. Empirical Validations of Theorem 3

In this subsection, we empirically validate the practical tightness of the upper bound in Theorem 3. Recall that Theorem 3 relies on Assumption 2, which assumes that the maximum edge-direction similarity $h(s)$ along the BTSP solution is bounded by a moderate constant $\beta$ with high probability. Our validation proceeds in two steps: (i) estimating $\beta$ from data for circulant digraphs, and (ii) examining the tightness of the resulting upper bound compared to both the achieved BCD and the worst-case upper bound in Theorem 2.

**Experimental Setups.** We exclude the ring graph from this analysis because BTSP-MSR employs a 2-approximation algorithm tailored to the ring graph, making the upper bounds trivial. We focus on the following three circulant digraphs: (n2) exponential, (n3) 1-peer exponential, and (n4) sparse exponential graphs. We generate geographically distributed physical nodes in the same setting as (T1). All reported results are averaged over 50 independent runs for each number of nodes $n \in \{10, \cdots, 100\}$.

**Estimation of $\beta$.** For each number of nodes $n$ and confidence level $\delta \in \{0.01, 0.05, 0.1\}$, we estimate the smallest value $\hat{\beta}(n, \delta)$ such that Assumption 2 holds empirically. Specifically, let $c^\star$ denote the multiplicative permutation that minimizes the MSR term, i.e., $c^\star = \arg\min_c g(\sigma_c S)$. For each skip width $s \in \bigcup_l \sigma_c S^{(l)}$, we compute the maximum edge-direction correlation $h(s)$ along the BTSP solution $\pi_1$. We then select $\hat{\beta}(n, \delta)$ as the empirical $(1 - \delta)$-quantile satisfying $h(s) \leq \hat{\beta}(n, \delta)$ simultaneously for all $s$.

Figure 15 reports the estimated $\hat{\beta}(n, \delta)$ for the three circulant digraphs. Across all cases, $\hat{\beta}(n, \delta)$ increases only mildly with $n$. Moreover, since $\beta$ affects the upper bound in Theorem 3 only through a square-root factor, it remains moderate even for a stringent confidence level $\delta = 0.01$. In all cases, $\hat{\beta}(n, \delta)$ remains sufficiently small, confirming that Assumption 2 is satisfied in practical NW scenarios.

**Tightness of the Practical Upper Bound.** Using the estimated $\hat{\beta}(n, \delta)$, we next evaluate the upper bound obtained by Theorem 3: $(B(\pi_1) - b)\sqrt{g(\sigma_c S) + \hat{\beta}(n, \delta)} + b$, and compare it with both the achieved BCD $\varphi(\pi^\star; \mathcal{E})$ and the worst-case bound $B(\pi_1)g(\sigma_c S)$ from Theorem 2. Figure 16 summarizes the results.

Across all three circulant digraphs and most numbers of $n$, the bound derived from Theorem 3 tightly upper-bounds the achieved BCD and closely follows its scaling behavior. In contrast, the worst-case bound from Theorem 2 grows with $n$ and is often one order of magnitude larger than the achieved BCD, rendering it overly conservative in practice. This difference is consistent with Figure 17, which shows that the MSR term $\min_c g(\sigma_c \mathcal{S})$ can remain small for sparse exponential graphs, whereas it grows with $n$ for the exponential graph, leading to loose worst-case bounds.

These observations demonstrate that the square-root dependence in Theorem 3, enabled by the bounded edge-direction similarity, captures the effective scaling of the BCD under realistic physical NWs.

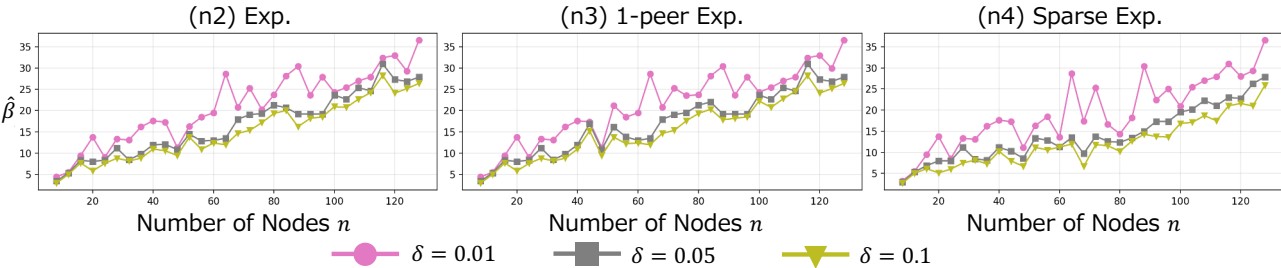

*Figure 15.* Estimated values of $\hat{\beta}(n, \delta)$ for three circulant digraphs: (n2) exponential, (n3) 1-peer exponential, and (n4) sparse exponential graphs. Each curve corresponds to a different confidence level $\delta$. Note that its effect on the upper bound in Theorem 3 enters only through a square-root term, and therefore does not dominate the resulting upper bound.

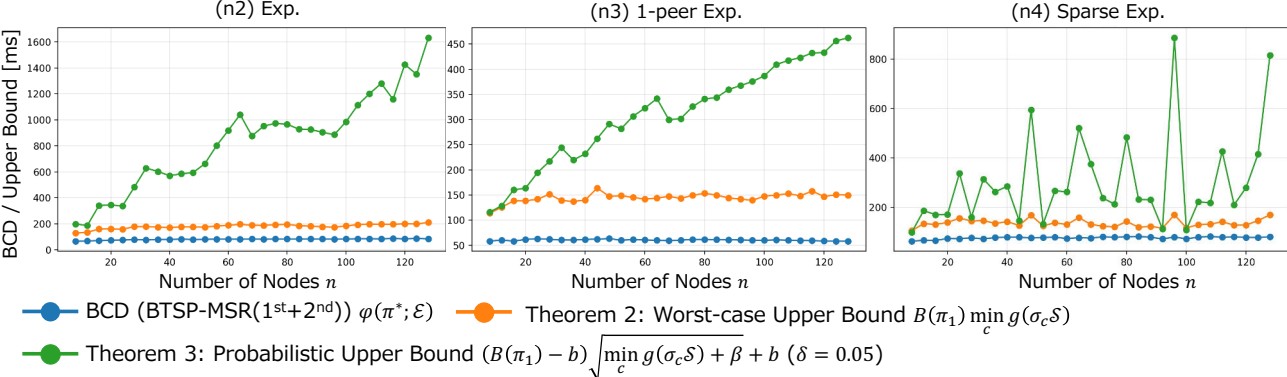

*Figure 16.* Comparison between the BCD and two theoretical upper bounds for three circulant digraphs: (n2) exponential, (n3) 1-peer exponential, and (n4) sparse exponential graphs.

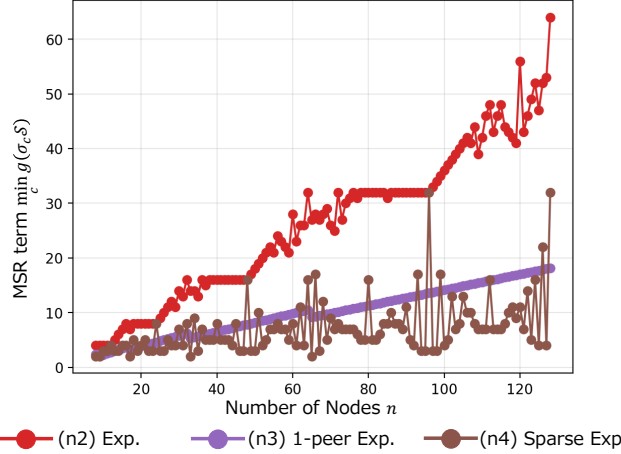

*Figure 17.* MSR term $\min_c g(\sigma_c \mathcal{S})$ for three circulant digraphs: (n2) exponential, (n3) 1-peer exponential, and (n4) sparse exponential graphs.

## E.6. Visualization of Resulting Node Assignment Examples

In (T1), we computed node assignments from four circulant digraphs (n1)–(n4) to simulated physical NWs over a wide range of network sizes $n$. As a representative example, Figures 18–21 visualize the assignments for $n = 57$. In each figure, the circulant digraph to be embedded is shown on the left, while the right panel illustrates the resulting assignment on a simulated physical NW, comparing the greedy solver with our proposed BTSP-MSR ($1^{st}+2^{nd}$). The link that attains the BCD is highlighted in red. Relative to the greedy solver, this red link is shorter under BTSP-MSR, indicating that the BCD is effectively reduced. In Figure 20, corresponding to the (n3) 1-peer exponential graph, there exists a time step $l$ at which BTSP-MSR yields a larger instantaneous BCD than the greedy solver. Nevertheless, since BCD is defined as the average of instantaneous BCDs across time steps as in (2), the overall BCD achieved by BTSP-MSR remains smaller than that of the greedy solver.

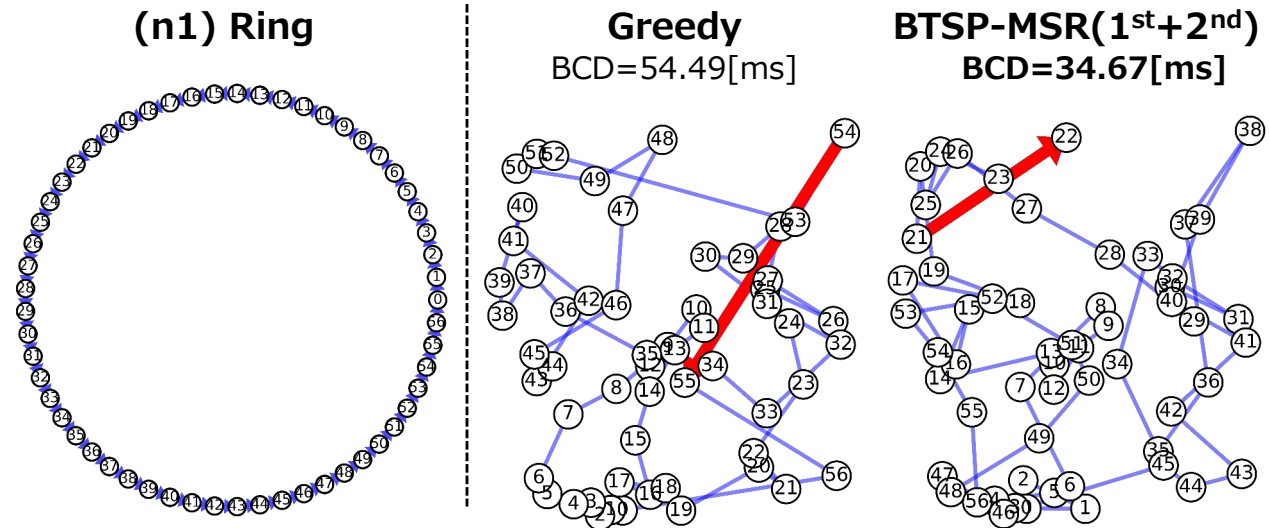

Figure 18. Visualization of node assignments for (n1) the ring graph with $n = 57$ under experiment (T1), along with the solutions of the BCD obtained by the greedy solver and BTSP-MSR(1st+2nd). The physical nodes are randomly placed within a square. Each red edge indicates the BCD under the given NW topology and assignment.

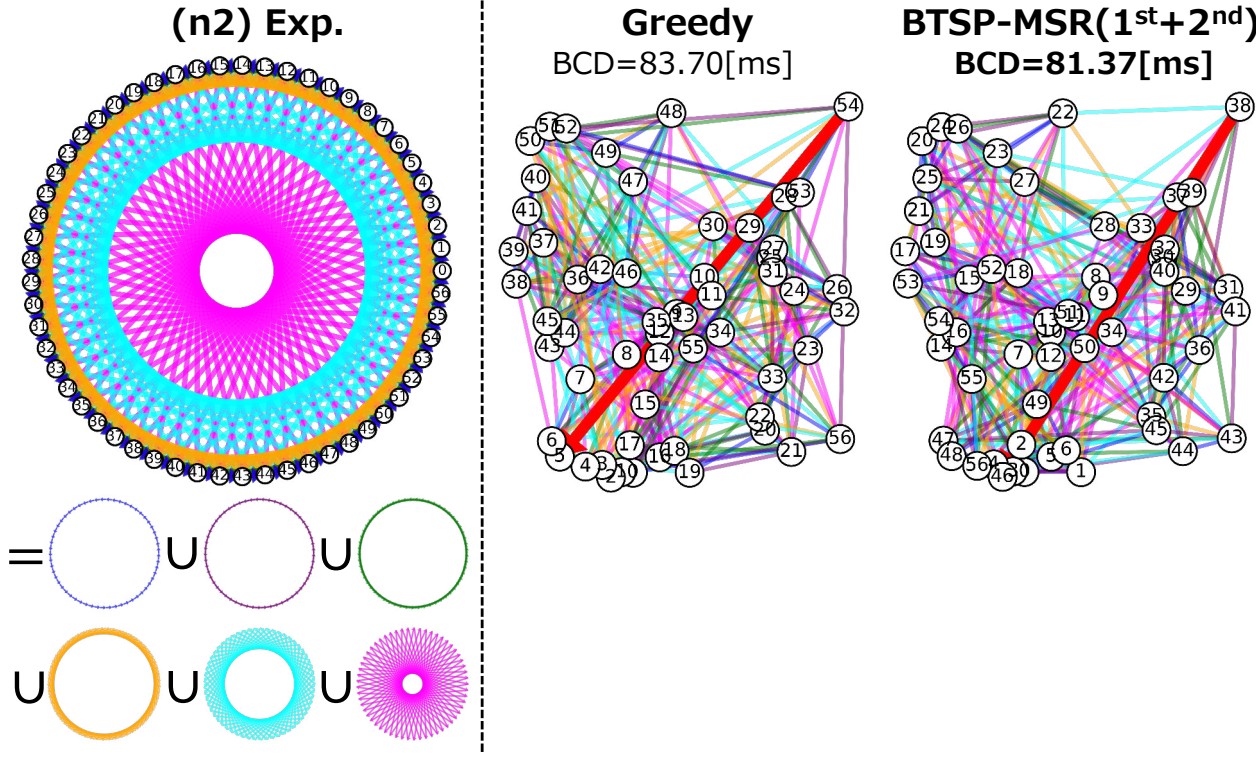

Figure 19. Visualization of node assignments for (n2) the exponential graph with $n = 57$ under experiment (T1), along with the solutions of the BCD obtained by the greedy solver and BTSP-MSR(1st+2nd). The physical nodes are randomly placed within a square. Each red edge indicates the BCD under the given NW topology and assignment.

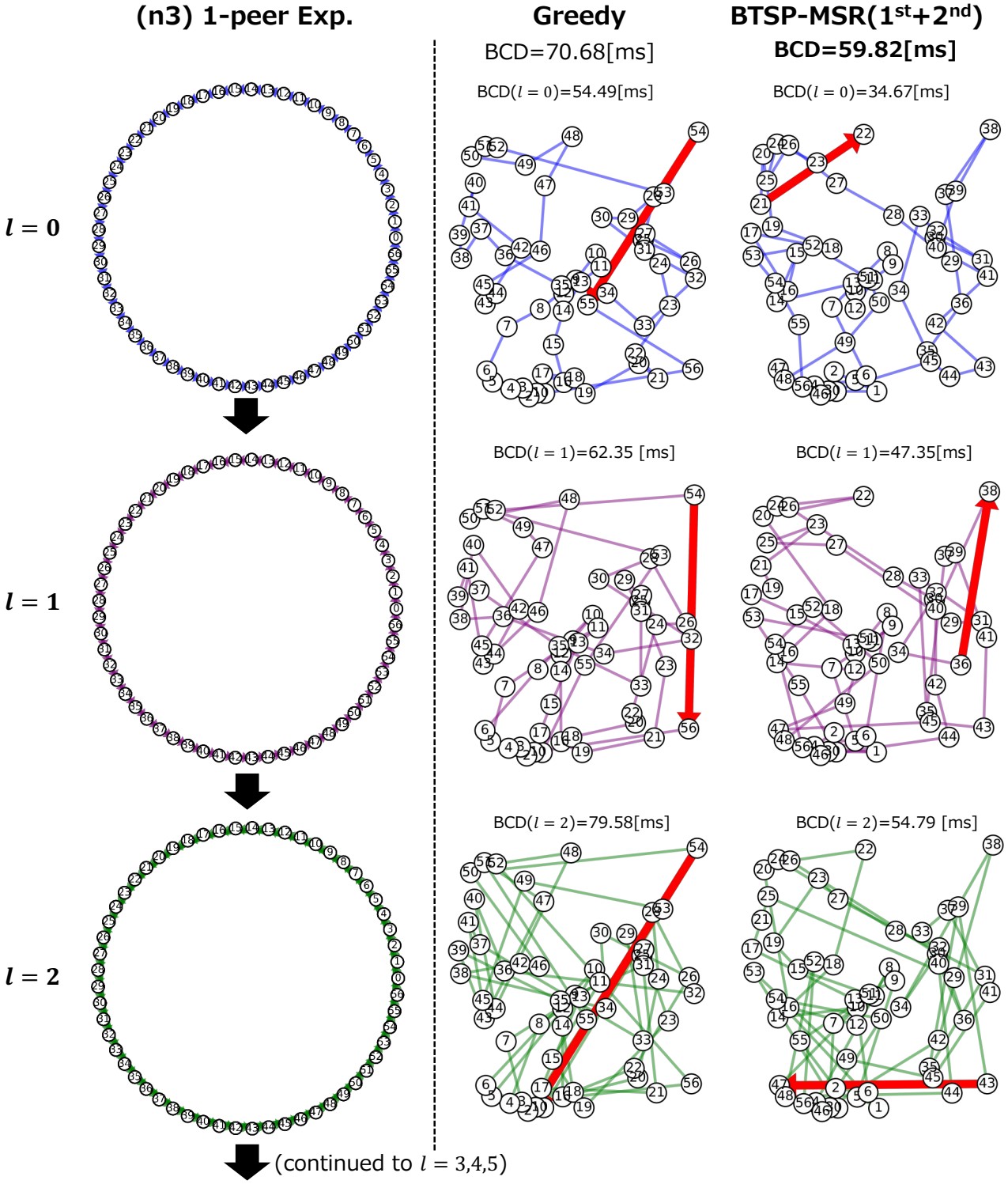

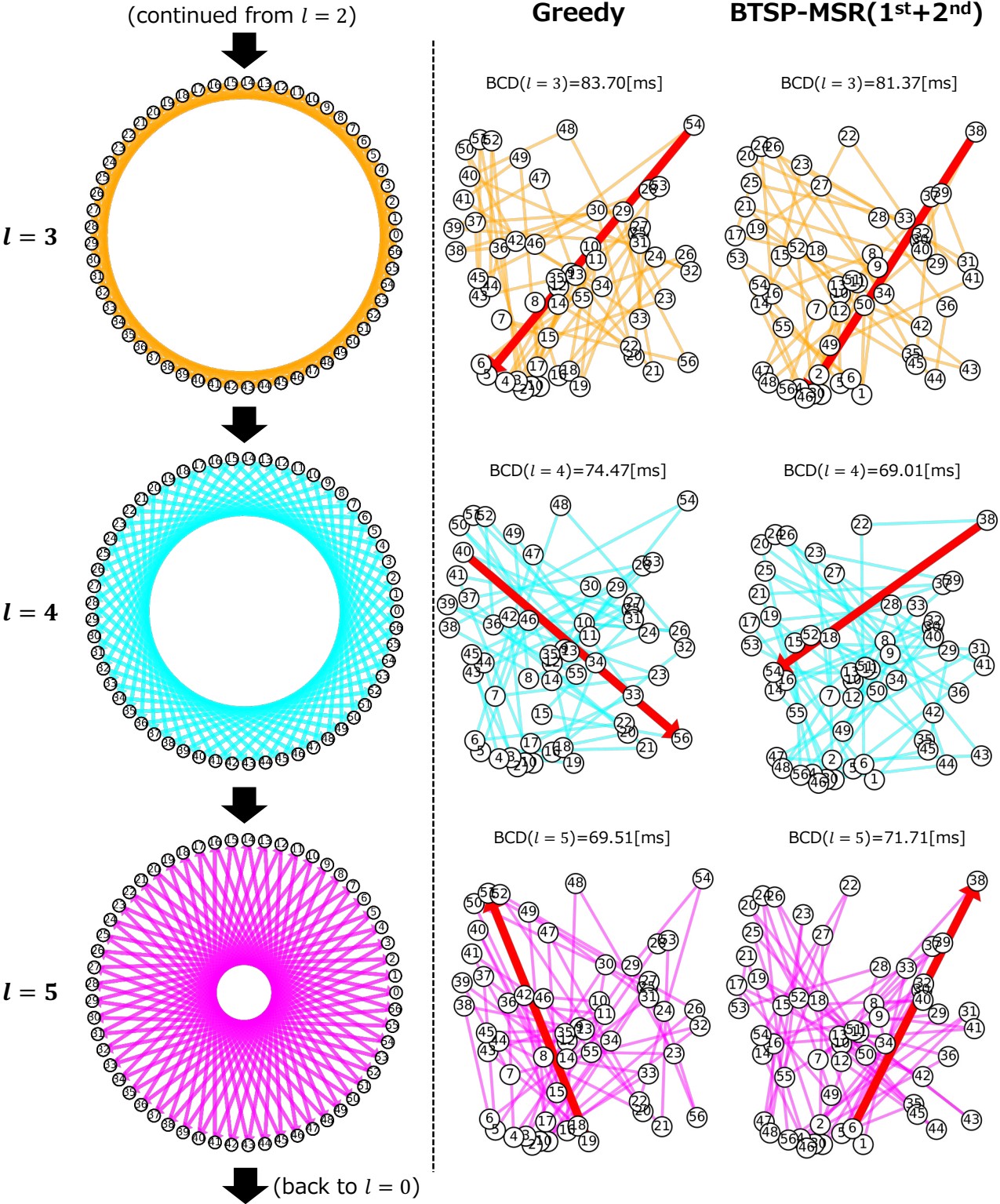

*Figure 20.* Visualization of node assignments for (n3) the 1-peer exponential graph with $n = 57$ under experiment (T1), along with the resulting BCD in each communication round $l \in \{0, 1, \dots, 5\}$ for the solutions obtained by the greedy solver and BTSP-MSR ($1^{\text{st}}+2^{\text{nd}}$). The red edges represent the bottleneck communication links in each round.

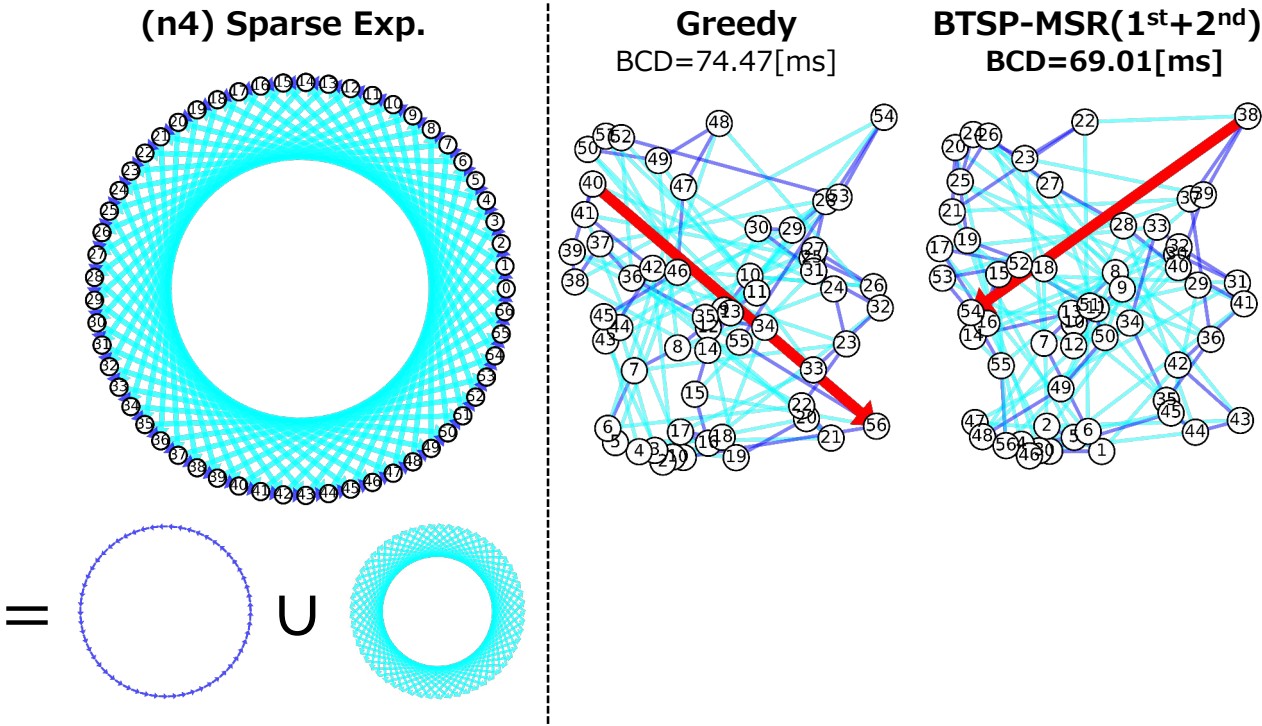

*Figure 21.* Visualization of node assignments for (n4) the sparse exponential graph with $n = 57$ under experiment (T1), along with the solutions of the BCD obtained by the greedy solver and BTSP-MSR($1^{st}$+$2^{nd}$). The physical nodes are randomly placed within a square. Each red edge indicates the BCD under the given NW topology and assignment.

# F. Experimental Setups and Results Regarding (T2) in Sec. 5.

This appendix section provides additional experimental details and analyses corresponding to (T2) in Sec. 5. We first describe the experimental setups and training configurations in Appendix F.1. We then report the best test accuracies in Appendix F.2, followed by additional decentralized learning results with Decentralized SCAFFOLD in Appendix F.3. Next, we analyze the relation between the test accuracies, NW topologies, and spectral gaps in Appendix F.4. We further investigate the applicability of the practical upper bound in Theorem 3 to real-world physical NW in Appendix F.5 and evaluate the robustness of BTSP-MSR under time-varying and asymmetric communication delays in Appendix F.6. Finally, we provide visualization results in Appendix F.7.

## F.1. Experimental Setups

**Data distribution.** We solved two image classification benchmark tests using CIFAR-10 and CIFAR-100. Their training dataset is divided into $n = 87$ nodes to be statistically heterogeneous, following a Dirichlet distribution with $\alpha = 0.1$ (Vogels et al., 2021). The resulting data distributions of the two datasets are illustrated in Figure 22.

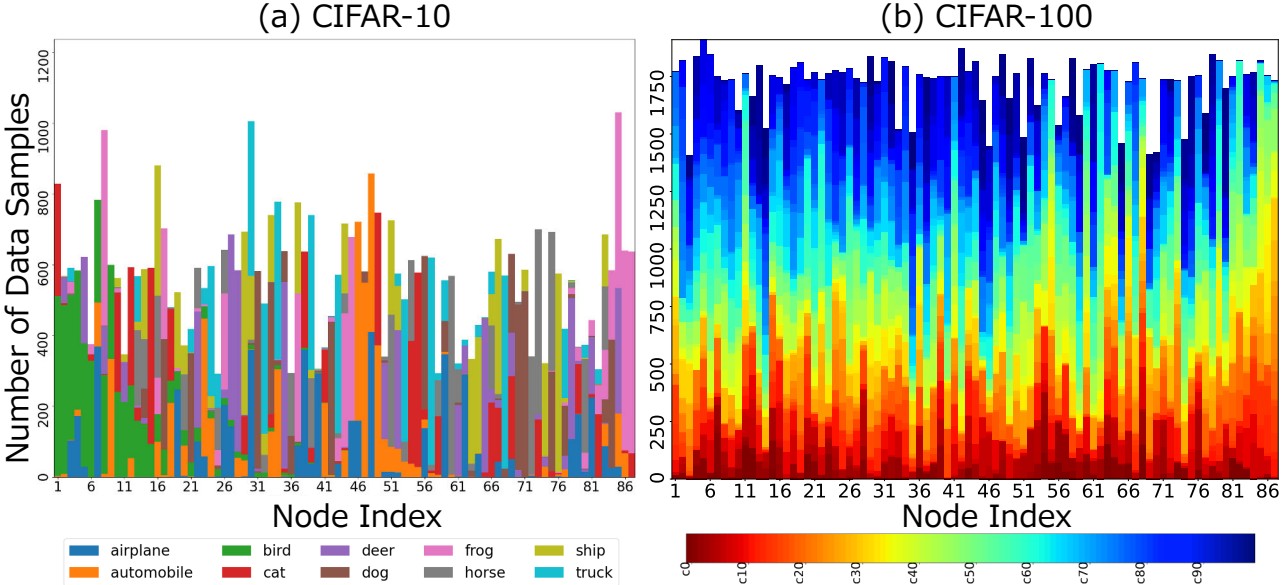

*Figure 22.* Heterogeneous data distributions of (a) CIFAR-10 and (b) CIFAR-100 over Ebone NW with $n = 87$ and Dir. $\alpha = 0.1$.

**Model Architectures and Hyperparameters.** We train LeNet-5 (LeCun et al., 2002) for CIFAR-10 classification task and ResNet-18 (He et al., 2016) for CIFAR-100 classification task. We used DSGD in (1) to train them, with the learning rate tuned in advance. To determine this value, we conducted a preliminary study of convergence curves on a ring graph using three candidate learning rates: $\{10^{-2}, 10^{-3}, 10^{-4}\}$. Among these, $10^{-3}$ was selected as it yielded the highest test accuracy. Other training settings are summarized in Table 2.

*Table 2.* Hyperparameters and settings for (T2) decentralized learning.

| Hyperparameters / settings | |
| --- | --- |
| Number of outer iterations (epochs) | 16,000 |
| Number of inner iterations (local updates) | 10 |
| Batch size | 128 |
| Learning rate | $1 \times 10^{-3}$ |
| Learning-rate scheduler | None |
| Warmup | None |
| Weight decay | 0.005 |
| Optimizer | DSGD |
| Data augmentation | RandomCrop, RandomHorizontalFlip, RandomErasing |

### F.2. Obtained Best Test Accuracy of Global Model

Table 3 reports the best test accuracies attained in (T2) on the two benchmark tests: (a) CIFAR-10 and (b) CIFAR-100 classification tasks, under several NW topologies. As shown in the table, (n2) the exponential and (n3) 1-peer exponential graphs optimized by BTSP-MSR ($1^{st}+2^{nd}$) consistently achieve high accuracy across both tasks. This trend is consistent with the theoretical expectation that NW topologies with larger spectral gaps enable faster information mixing and thus improve convergence under heterogeneous communication delays, leading to better final model performance. In contrast, MATCHA+ favors more localized communication and yields lower accuracies, suggesting that insufficient global mixing can hinder convergence in decentralized learning. In the next subsection, we empirically substantiate this interpretation by estimating the spectral gap of each NW topology.

*Table 3.* Best test accuracy in Experiment (T2) for (a) CIFAR-10 and (b) CIFAR-100. Higher accuracy is better.

| NW topology (Solver) | (a) CIFAR-10 Acc. | (b) CIFAR-100 Acc. |
| --- | --- | --- |
| MATCHA+ | 0.567 | 0.662 |
| (n1) Ring (BTSP-MSR(1st+2nd)) | 0.561 | 0.656 |
| (n2) Exp. (BTSP-MSR(1st+2nd)) | 0.589 | 0.714 |
| (n3) 1-peer Exp. (BTSP-MSR(1st+2nd)) | 0.579 | 0.715 |
| (n4) Sparse Exp. (BTSP-MSR(1st+2nd)) | 0.554 | 0.718 |

### F.3. Additional Results Using Decentralized SCAFFOLD

In the decentralized learning experiment (T2) in Sec. 5, we used DSGD as the default decentralized learning algorithm, as described in Appendix F.1. To confirm that BTSP-MSR is also effective with decentralized learning methods other than DSGD, we additionally conducted experiments using Decentralized SCAFFOLD (Liu et al., 2021) on CIFAR-10 and CIFAR-100 under the same physical NW and NW topologies as in (T2).

For CIFAR-10, we evaluated all NW topologies considered in (T2). For CIFAR-100, due to computational resource limitations, we fixed the learning rate to $10^{-3}$ and evaluated only the 1-peer exponential and sparse exponential graphs, along with MATCHA+. We also capped the total number of update iterations. Since MATCHA+ has a smaller BCD than the other topologies, this fixed update-iteration budget corresponds to a shorter elapsed-time horizon for MATCHA+.

Figure 23 presents the results using the global model parameter on the two datasets. Similar convergence trends are observed even under this different decentralized learning algorithm; in particular, combining BTSP-MSR with advanced NW topologies with large spectral gaps remains effective, supporting our strategy of reducing BCD while retaining communication-efficient NW topologies.

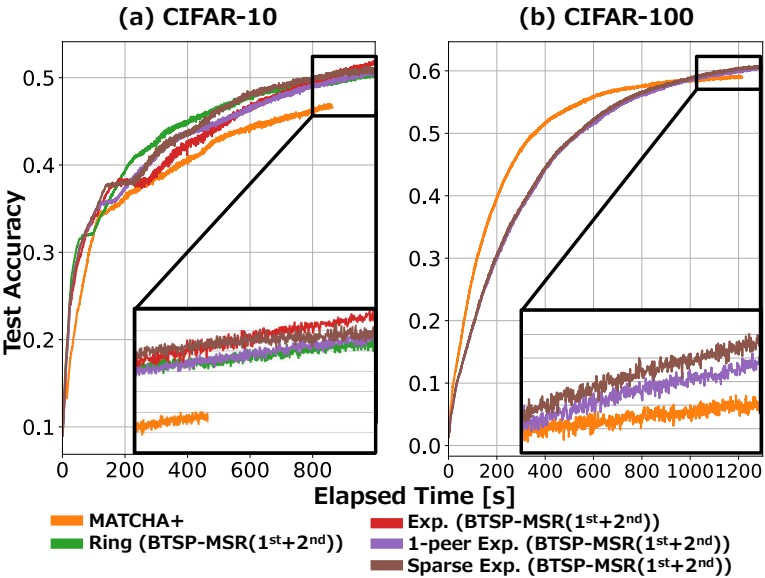

*Figure 23.* Decentralized learning results using Decentralized SCAFFOLD on the (a) CIFAR-10 and (b) CIFAR-100 datasets. Accounting for communication delays, the horizontal axis represents elapsed time. Since MATCHA+ has a smaller BCD than the other topologies, the same number of update iterations corresponds to a shorter elapsed-time horizon. Due to computational resource limitations, we capped the total number of update iterations, and for CIFAR-100, we fixed the learning rate to $10^{-3}$ and evaluated only the 1-peer exponential and sparse exponential graphs.

## F.4. Estimation of Spectral Gaps

In this subsection, we empirically estimated the spectral gap to isolate and assess the impact of the NW topologies used in (T2). In the decentralized learning experiments of (T2), the convergence curves using test accuracy reflect a mix of various factors–including dataset characteristics, local update rules, and stochastic noise–making it difficult to attribute performance differences solely to NW topology. To estimate the spectral gap defined in Appendix B, Assumption 3, we computed the consensus error, which measures the deviation of each node's local average from the global average, thereby providing a NW topology-focused evaluation independent of other confounding effects.

For this aim, we begin by randomly selecting scalars $x_i^{(0)} \in \mathbb{R}$ ($i = 0, \cdots, n-1$) drawn from the uniform distribution $U(0, 1)$. Then, we evaluate the decrease of the consensus error $e^{(l)} = \frac{1}{n} \sum_{i=0}^{n-1} (x_i^{(l)} - \bar{x})$ with updating $x_i^{(l+1)} \leftarrow \sum_{j=0}^{n-1} W_{ij}^{(l)} x_i^{(l)}$ where $\bar{x} = \frac{1}{n} \sum_{i=0}^{n-1} x_i^{(0)}$. By observing how quickly the consensus error converges to zero, we can evaluate the effectiveness of the NW topology for averaging; faster convergence indicates better averaging performance. To reflect the effect of heterogeneous communication delays, we used the Ebone NW, the same physical NW in (T2), and we compared the consensus error for each NW topology by taking the BCD into account.

Figure 24 shows the consensus errors of several NW topologies with Ebone NW ($n = 87$). The slope of the consensus error decay in Figure 24 can be used to estimate the spectral gap of the underlying NW topology. Since the expected spectral gap $\lambda$ is defined as a constant satisfying the inequality

$$\mathbb{E}\big[\|e^{(l+1)}\|^2\big] \leq (1 - \lambda) \|e^{(l)}\|_F^2,$$

where the consensus error $e^{(l)}$ decreases approximately at a geometric rate, $e^{(l)} \leq (1 - \lambda)^l e^{(0)}$. Therefore, the expected spectral gap can be empirically estimated from the decay rate of the consensus error, and the slope of the curves in Figure 24 directly reflects the effective averaging capability of each NW topology. We report the estimated spectral gap $\lambda$ in Table 4 from the empirical error sequence by the geometric mean of the ratios $e^{(l+1)}/e^{(l)}$ over the last 25 iterations.

Among the NW topologies, the exponential graph achieves the fastest consensus, which is consistent with information in Table 1. In contrast, despite having a relatively small BCD, MATCHA+ exhibits slower convergence in this experiment. This empirical result suggests that the expected spectral gap of MATCHA+ may be smaller than that of the exponential graph and the 1-peer exponential graph.

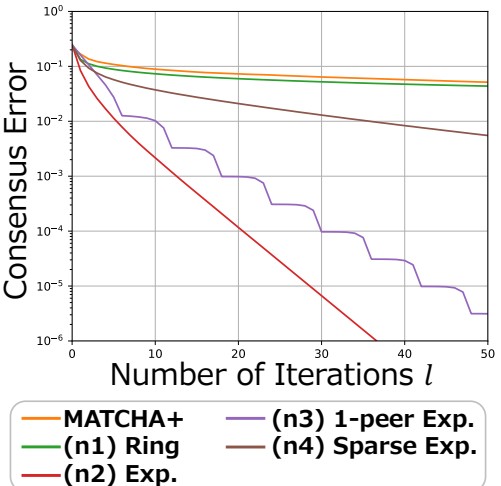

*Figure 24.* Comparison of consensus error of several NW topologies with Ebone NW ($n = 87$), used in (T2)

| | |
|---|---|
| MATCHA+ | 0.0075 |
| (n1) Ring | 0.0096 |
| (n2) Exp. | 0.2488 |
| (n3) 1-peer Exp. | 0.2268 |
| (n4) Sparse Exp. | 0.0427 |

*Table 4.* Estimated spectral gaps

### F.5. Applicability of Tighter Upper Bound in Theorem 3 to Real-world NW Scenario in (T2)

Let us recall that Theorem 3 is derived under Assumptions 1 and 2. However, Assumption 1 does not hold for the real-world physical network (NW) used in (T2). In (T2), two sites $p$ and $q$ are not necessarily directly connected, and we define $\mathrm{dis}(p,q)$ as their shortest-path distance. Therefore, Theorem 3 is not rigorously applicable in this setting; nevertheless, we argue that it can serve as a good approximation even for the physical NW considered in (T2).

To support this claim, we embed the $n$ node positions into a high-dimensional Euclidean space so that communication delays among the nodes are well-approximated by Euclidean distances between the embedded nodes. Let $Z_0, \ldots, Z_{n-1}$ denote the embedded node positions. By Bourgain's theorem (Bourgain, 1985), there exist points $Z_0, \ldots, Z_{n-1}$ such that

$$d(i,j) \approx \|Z_i - Z_j\|.$$

This suggests that, for (T2), Assumption 1 can be satisfied approximately, effectively with $\kappa = 1$ and $b = 0$ in the embedded high-dimensional space.

We empirically validate this approximation by embedding the nodes of the Ebone NW into a high-dimensional Euclidean space. Using a 10-dimensional embedding as an example, the average error between the original communication delays and their Euclidean approximations is 1.73 [ms], which is sufficiently small. This result supports using Assumption 1 for (T2) as a practical approximation.

Based on this embedding, we further compute the tighter upper bound for (T2), with results summarized in Table 5. Since the physical node locations in (T2) are fixed rather than randomly sampled, Assumption 2 is not directly applicable. Accordingly, instead of estimating $\beta$ from data as in Appendix E.5, we report the resulting tighter upper bounds for several representative values of $\beta$.

As shown in Table 5, across all three circulant NW topologies, the tighter bounds from Theorem 3 are substantially smaller than the worst-case bound from Theorem 2, even for relatively large values of $\beta$. This indicates that the tighter upper bound in Theorem 3 remains informative for (T2). In particular, for the exponential graph, the discrepancy between the two upper bounds is substantial, in line with the experimental results in Appendix E.5.

*Table 5.* Averaged BCD and upper bounds in (T2) under different NW topologies.

| NW topology | BCD [ms] | Tighter UB (Theorem 3: Approximated) | | | Worst-case UB (Theorem 2) |
|---|---|---|---|---|---|
| | | $\beta = 20$ | $\beta = 30$ | $\beta = 40$ | |
| (n2) Exp. | 85.08 | 491.2 | 536.0 | 577.8 | 2178.6 |
| (n3) 1-peer Exp. | 79.79 | 387.2 | 442.6 | 492.0 | 836.6 |
| (n4) Sparse Exp. | 76.49 | 360.2 | 419.7 | 471.8 | 544.6 |

## F.6. Robustness of BTSP-MSR to Time-Varying and Asymmetric Perturbations of Communication Delays

In this subsection, we investigate the robustness of the node assignments by BTSP-MSR ($1^{st}+2^{nd}$) under time-varying and asymmetric perturbations of the communication delay function $d$. Since communication delay in physical NWs inevitably fluctuates due to congestion, routing changes, or measurement noise, it is important to verify that the obtained node assignment does not critically depend on the precise values of the original communication delay function $d$.

To evaluate the robustness of BTSP-MSR to such a perturbation, we consider perturbed communication delay function $d^r$ as follows. Let $\pi^*$ denote the node assignment obtained from BTSP-MSR ($1^{st}+2^{nd}$) under the original communication delay function $d$. We construct a perturbed communication delay function $d^r$ by slowing down the original delays, where each perturbed communication delay $d_{ij}^r$ is sampled independently from the uniform distribution $U(d_{ij}, r \cdot d_{ij})$, where $r > 1$ is a perturbation strength. This procedure introduces both asymmetric ($d^r(i,j) \neq d^r(j,i)$) and time-varying communication delay function.

To investigate the robustness of BTSP-MSR, Figure 25 plots two quantities on the Ebone NW: (i) the original BCD and (ii) the BCD with the node assignment obtained from BTSP-MSR ($1^{st}+2^{nd}$), with respect to the perturbation strength $r \in [1, 3]$.

The results show that increasing $r$ leads to a gradual increase in the BCD, while sudden degradation is not observed when the communication delay function becomes asymmetric or time-varying. Moreover, across the entire range of perturbation strengths, BTSP-MSR consistently achieves lower BCDs than the greedy solver, indicating that its advantage is preserved even under asymmetric and time-varying communication delay function.

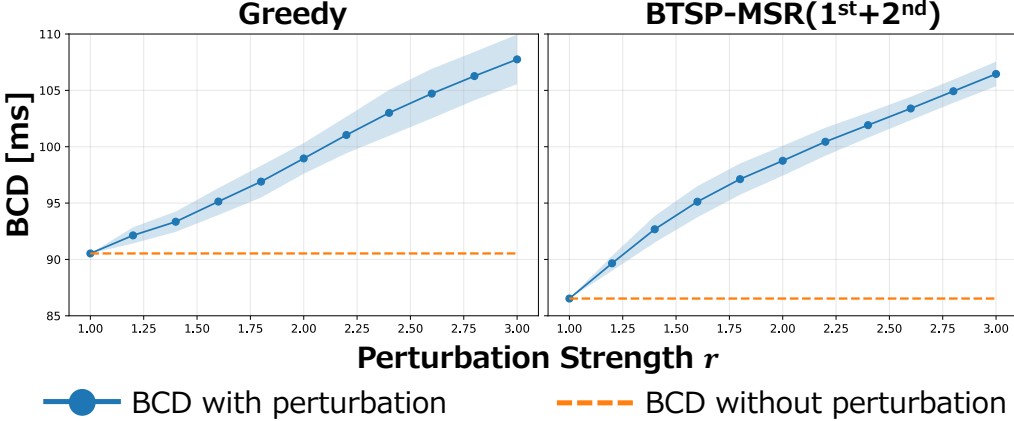

*Figure 25.* Robustness of BTSP-MSR($1^{st}+2^{nd}$) and the greedy solver on the exponential graph from slowing down the communication delay on Ebone NW ($n = 87$), where the perturbed communication delay $d_{ij}^r$ is sampled from the uniform distribution $U(d_{ij}, rd_{ij})$. Each result is averaged over ten independent runs.

## F.7. Visualization of BCD

In (T2), we computed node assignments from four circulant digraphs (n1)–(n4) onto a real physical NW (the Ebone network) with $n = 87$. Figure 26 visualizes the resulting bottleneck communication-delay (BCD) paths, highlighted in red. We show only these BCD paths because plotting the full assignment—analogous to Figures 18–21—would clutter the figure and obscure the main comparison. Because the Ebone network is not fully connected, the communication delay between two non-adjacent physical nodes is defined as the delay of the shortest-delay path between them. We also note that several physical nodes are co-located, reflecting realistic deployments (e.g., multiple compute nodes within the same data center). Compared with the greedy solver, our proposed BTSP-MSR ($1^{st}+2^{nd}$) tends to avoid geographically long communication routes. Consequently, it consistently yields shorter bottleneck paths and achieves lower BCD values, as reported in Figure 7.

**Physical NW**

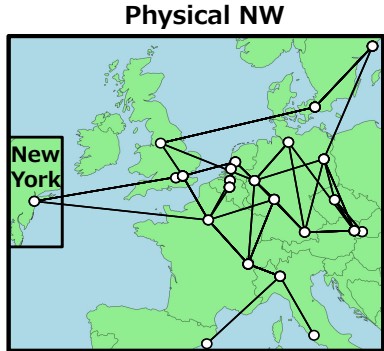

*Figure 26.* Visualization of Ebone NW with $n = 87$ nodes. Some physical nodes are co-located in the same position. The physical NW is not fully connected, and when nodes $p$ and $q$ are not directly connected, $d(p, q)$ is defined as the total delay along the shortest-delay path between them. Accordingly, some edges in the given NW topology correspond to paths over this NW.

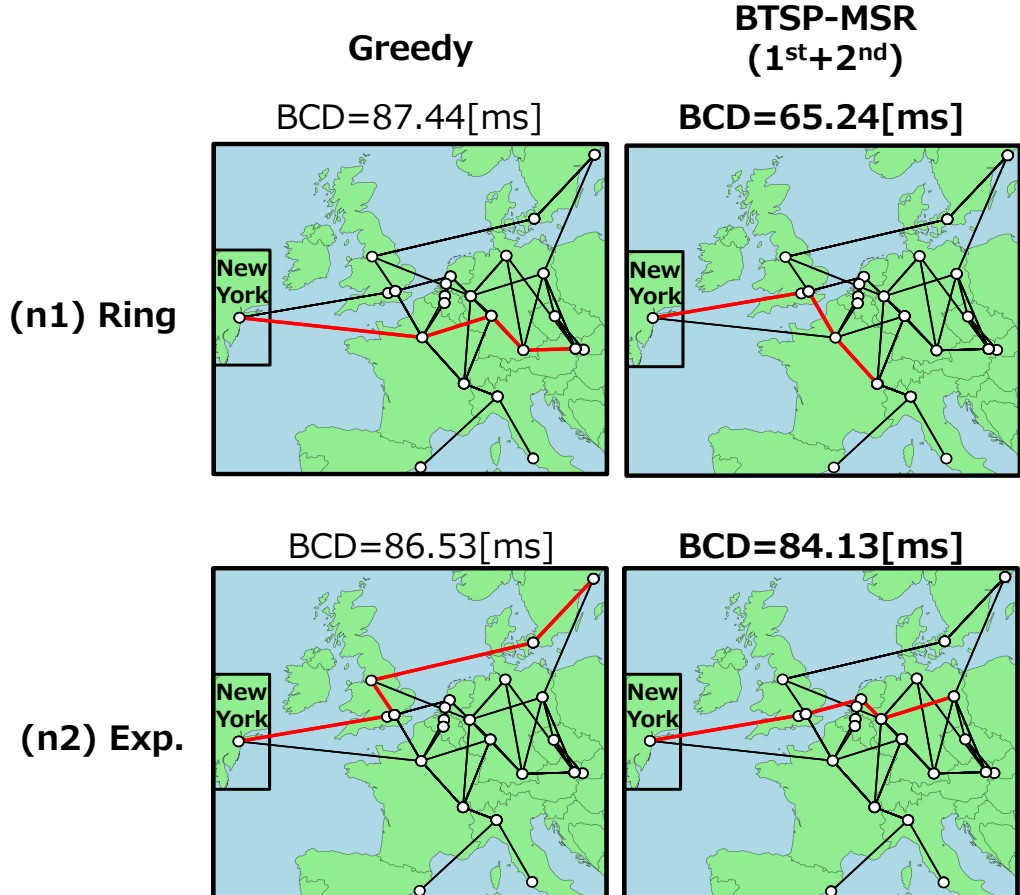

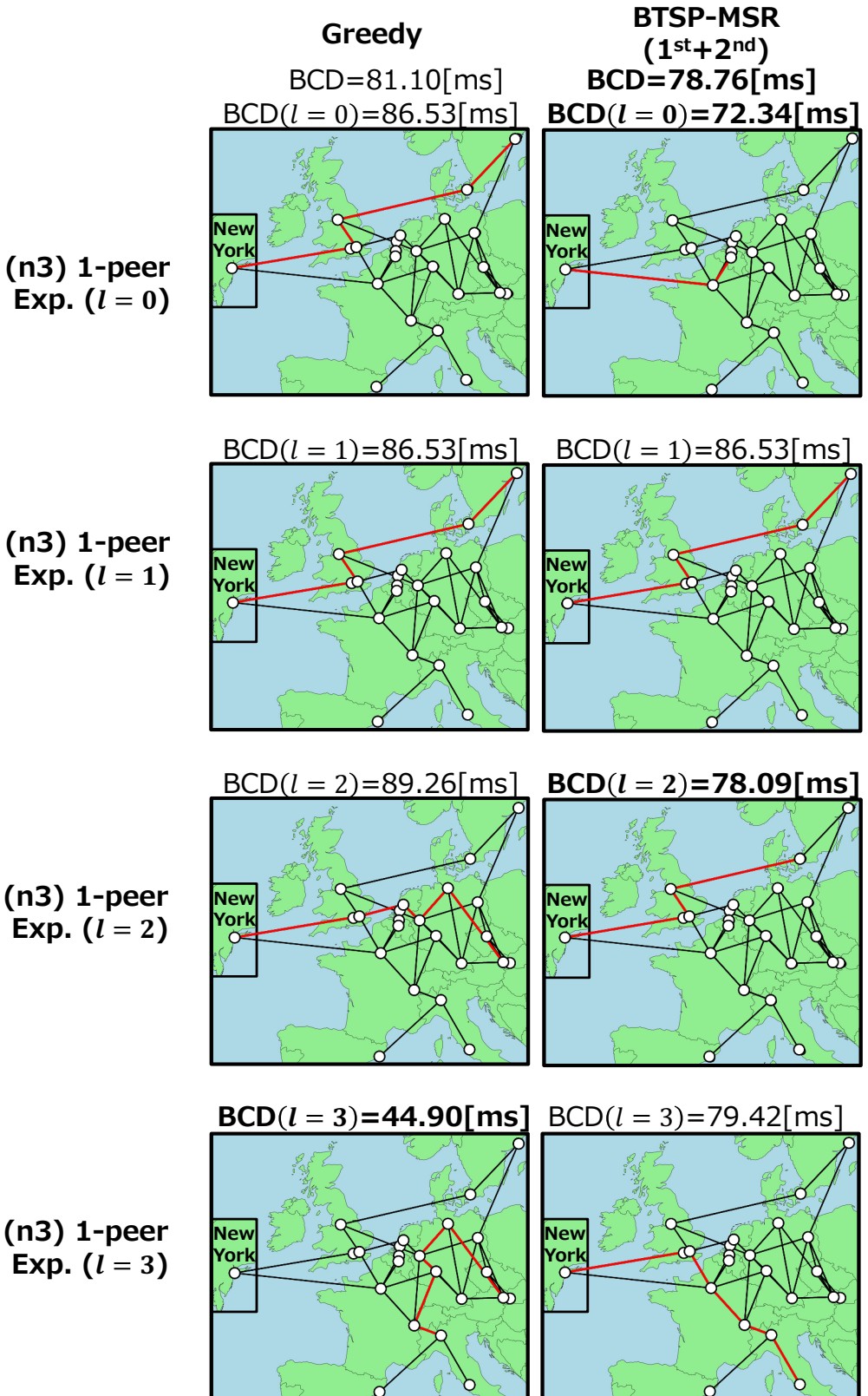

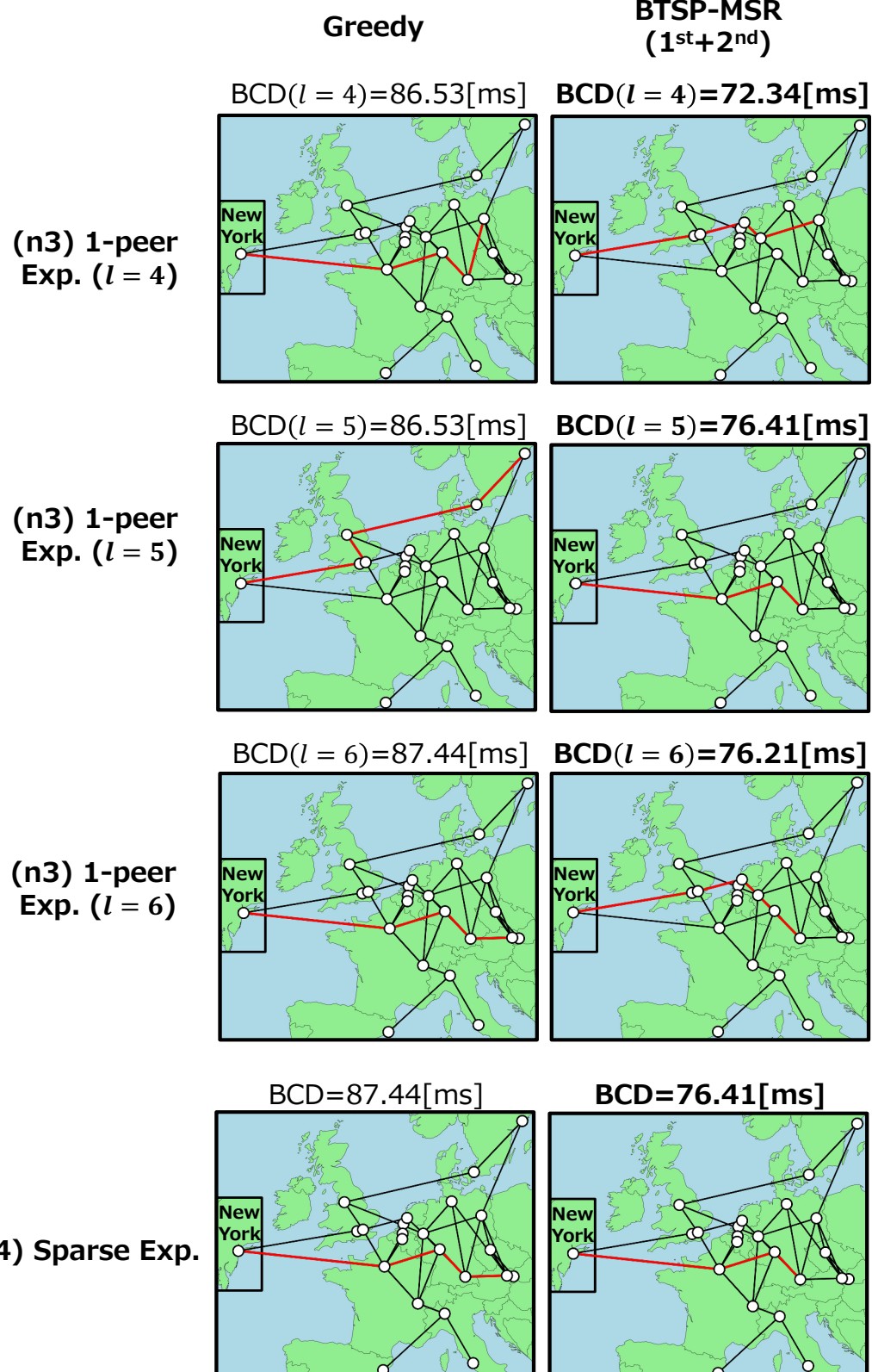

*Figure 27.* Visualization of BCDs on the Ebone NW for the ring, exponential, 1-peer exponential and sparse exponential graphs, and node assignment solvers (the greedy solver and BTSP-MSR (1st+2nd)). Since the bottleneck communication occurred between physical nodes that are not directly connected, the BCDs are represented as red paths. Compared to the greedy solver, BTSP-MSR (1st+2nd) assigns nodes to avoid communication between geographically distant nodes for most cases, effectively reducing the BCD.

