# OpenReview forum: "Bottleneck Communication Delay Minimization for Communication-Efficient Decentralized Learning"
_ICML.cc/2026/Conference — ICML 2026 regular_

### Official Review · Reviewer_EBs7 · 2026-03-05

**Soundness:** 3
**Presentation:** 3
**Significance:** 3
**Originality:** 3
**Overall Recommendation:** 4
**Confidence:** 4

**Summary:**

This paper considers how to assign virtual nodes on a network topology to physical computing nodes, in order to minimize the bottleneck communication delay (BCD). The author(s) formulate it as a node assignment problem, which is shown to be NP-hard. Moreover, approximation methods are given to approximately solve the problem. Experiments are provided to demonstrate that the proposed methods can give smaller BCDs than random method and greedy method, yielding faster convergence of a decentralized learning on the network with better assignments.

**Compliance With Llm Reviewing Policy:**

Affirmed.

**Final Justification:**

The authors have addressed my main concerns.

**Key Questions For Authors:**

1. Can the author(s) clarify why the network should be fixed first?

2. Can the design of the mixing matrix $W$ be incorporated with the design of network and/or node assignment?

3. In problem (3), should a communication restriction be included in the formula?

**Limitations:**

See "Strengths And Weaknesses"

**Strengths And Weaknesses:**

Strengths:

1. The experimental results support the claimed results.

2. The motivation of minimizing BCD is clear.

Weaknesses:

1. Though Bottle Communication Delay can affect the convergence speed of a decentralized algorithm, there are a lot of other factors affecting the speed, such as the spectral gap as the paper mentions. It is unclear how important reducing BCD is compared to others factors (such as designing the mixing matrix $W$).

2. The paper assumes that the communication network is fixed first and then performs the node assignment. But it is not clear why the network topology should be fixed. It would be more convincing to optimize the communication network given certain restrictions on the communication (like communication delay).

3. Because the proposed methods are heuristic, more experiments would be needed to convince the validity of the idea. It is not clear what decentralized learning method was tested in the experiments and what the mixing matrix is. Different methods may be affected by the communication network differently.

4. The paper appears to be too compact. It looks like that the author(s) intentionally squeezed some space in the paper and use smaller fonts in some math equations.

---

> ### Author Rebuttal · Authors · 2026-03-31
>
> We sincerely thank you for your thoughtful review and valuable feedback. Due to the space limitation, we were not able to address all points in full detail, but we would be happy to provide further clarification if needed. __We hope that our responses have addressed your main concerns, and we would greatly appreciate your consideration in updating your evaluation if appropriate__.
>
> # W1: Importance of reducing BCD compared to other factors.
>
> Our strategy aims to provide BCD reduction methods for advanced NW topologies with large spectral gaps (e.g., exponential and 1-peer exponential graphs). We interpret your feedback as indicating that the importance of this strategy may not be supported by clear evidence in the current version.
>
> __One piece of evidence supporting the effectiveness of our strategy is the analytically estimated time required to reach a target level, which depends primarily on the spectral gap of the graph and secondarily on the BCD time.__
> Theorem 4 in Appendix A characterizes the convergence rate of the decentralized learning algorithm (DSGD). Specifically, it provides an upper bound on the number of communication rounds $R$ required for the squared gradient norm to reach a target level $\epsilon$, in terms of the spectral gap $\lambda$, the number of mixing matrices $\tau$, the data heterogeneity $\zeta$, variance of the stochastic gradient noise $\sigma^2$, Lipschitz smoothness $L$, number of nodes $n$, and distance $f_0$. Then, the analytically estimated time required to reach a target level $\epsilon$ is obtained by
>
> $$
> T_{est} = T_{BCD} \cdot R =O\left(\left(\frac{\sigma^2}{n\epsilon^2} + \frac{\zeta\tau+\sigma\sqrt{\lambda\tau}}{\lambda\epsilon^{3/2}} + \frac{\tau}{\lambda\epsilon} \right) \cdot f_0 L T_{BCD}\right)
> $$
>
> Since hyperparameters (e.g., $\zeta, \sigma^2, L, f_0, n$) depend on the problem setting and are not directly controllable, it is reasonable to simplify the discussion by fixing them (e.g., $n=87, \sigma^2 = 1, \zeta = 1, f_0 = 1, L = 1, \epsilon = 10^{-2}$). Under this setting, $T_{est}$ can be expressed as a function of the per-round spectral gap $\lambda / \tau$ and the BCD time $T_{BCD}$, illustrated as following figure: __https://anonymous.4open.science/r/ICML2026_rebuttal-F654/estimated_runtime.png__
>
> From this figure, we observe that __the total time__ $T_{est}$ __can be reduced primarily by improving the spectral gap of the NW topology and secondarily by reducing the BCD__. This supports our strategy of developing BCD reduction methods for advanced NW topologies with large spectral gaps (e.g., exponential and 1-peer exponential graphs). We will incorporate this discussion in the revised version.
>
> # W2, Q1: Why the network topology should be fixed?
>
> As discussed in W1, the total time $T_{est}$ is primarily affected by the spectral gap. Thus, we adopt a strategy that first selects advanced NW topologies with large spectral gaps and provides their BCD reduction method.
>
> On the other hand, one method that prioritizes BCD reduction at the expense of spectral gap is MATCHA [Wang et al., 2022]. However, in our experiments (T2 in Sec. 5), we empirically observe that our strategy—selecting NW topologies with large spectral gaps and then reducing BCD—leads to better performance in decentralized learning.
>
> # W3: Concerns about heuristic nature of proposed method, and unclear descriptions regarding experimental setups.
>
> First, our question is: __what is meant by "heuristic nature of the proposed method"?__. For our BTSP-MSR, __we provide theoretical analyses in Sec. 4.3__ and their empirical evaluations in Appendix E.3. We confirmed that the bound given by Theorem 3 is tight in practice (see Figure 13).
>
> Second, in the experiment (T2) of Sec. 5, we used DSGD as the decentralized learning algorithm, clarified in Appendix F.1. Additionally, we conducted experiments using another decentralized learning algorithm (Decentralized SCAFFOLD [Xing et al., 2020]). We report results on CIFAR-10 for NW topologies in (T2): __https://anonymous.4open.science/r/ICML2026_rebuttal-F654/CIFAR10_scaffold.png__.
>
> __Even under different decentralized learning algorithms, similar convergence trends are observed; namely, our strategy of providing BCD reduction methods for NW topologies with large spectral gaps remains effective__. We will include additional results for CIFAR-100 in the revised version.
>
> # Q3: Should a communication restriction be included in the formula (3)?
>
> In the optimization problem (3), we assume that any pair of nodes can communicate.
> Handling cases where communication is restricted to specific pairs is future work. In our setting, we assume that communication is possible via intermediate nodes, where the communication delay is computed as the sum of delays.
> In our experiments (T2), we use the Ebone NW, where all-to-all direct communication is not available, and such multi-hop communication delays are taken into account, and visualized in Figure 22-23 (Appendix F.6).

---

> > ### Author Rebuttal · Reviewer_EBs7 · 2026-04-01
> >
> > I appreciate the authors' responses. They did address some of my concerns/questions. But some concerns remain.
> >
> > 1. The authors acknowledged that spectral gap of the network is the first factor impacting $T_{est}$. Can the authors elaborate more on how to incorporate the design of mixing matrix in their framework or explain why it cannot be incorporated?
> >
> > 2. By "heuristic nature of the proposed method", I mean that the formulated problem (3) is NP-hard, so no polynomial-time algorithm can be found. Therefore, the proposed method for solving (3) can only be heuristic. It is good to have an approximation ratio guarantee.
> >
> > I hope the authors can address my first question above.
> >
> > **Acknowledgement to follow up responses**
> >
> > I thank the authors' further clarification. My concerns are now fully resolved. I increased my score to 4 now.

---

> > > ### Author Response · Authors · 2026-04-02
> > >
> > > Thank you for the opportunity to clarify these points.
> > >
> > > 1: Your suggestion of jointly optimizing the mixing matrix is certainly interesting, but also substantially more complex.
> > > As you pointed out, "spectral gap of the network is the first factor impacting", followed by the BCD time. Given this observation, we believe there are two reasonable strategies for reducing the total training time:
> > >
> > > __(strategy 1) Developing a BCD reduction method under advanced NW topologies that already provide a larger spectral gap.__
> > >
> > > __(strategy 2) Jointly optimizing the NW topology and the BCD reduction method to minimize the overall training time.__
> > >
> > > Among these two directions, __strategy 1 is more tractable, and this is the setting we focus on in this paper.__ While advanced NW topologies (e.g., exponential and 1-peer exponential graphs) have been studied from the perspective of spectral gap, BCD minimization method over such NW topologies has been unexplored, which motivates our focus on strategy 1.
> > >
> > > In contrast, __your suggestion would correspond to strategy 2__, which is certainly interesting, but also substantially more complex and challenging. This is because our analytically estimated training time (see our response in W1) is a __nonlinear optimization problem__ with respect to spectral gap $\lambda$ and BCD time $T_{BCD}$ as follows, even when other hyperparameters are fixed (e.g., $\sigma^2, L, \zeta$):
> > >
> > > $$T_{est} = T_{BCD} \cdot R = O\left( \left( \frac{\sigma^2}{n\epsilon^2} + \frac{\zeta\tau + \sigma\sqrt{\lambda\tau}}{\lambda \epsilon^{3/2}} + \frac{\tau}{\lambda \epsilon} \right) \cdot f_0 L  T_{BCD} \right).$$
> > >
> > > In addition, both designing NW topologies (to increase $\lambda$) and BCD reduction (to reduce $T_{BCD}$) can be combinatorial optimization problem. Thus, __strategy 2 is nonlinear combinatorial optimization problem, which is substantially more complex and challenging.__ We therefore view this as an important direction for future work.
> > >
> > >
> > >
> > > 2: Thank you for clarifying what you meant by the “heuristic nature of the proposed method.” We understand your point. As you noted, "no polynomial-time algorithm can be found" for the formulated problem (3), it is natural to develop an approximation method to address it. We also appreciate your positive comment "it is good to have an approximation ratio guarantee".

---

### Official Review · Reviewer_7MoZ · 2026-03-10

**Soundness:** 3
**Presentation:** 3
**Significance:** 2
**Originality:** 1
**Overall Recommendation:** 4
**Confidence:** 3

**Summary:**

This paper investigates the problem of optimizing networks to minimize communication delays in decentralized learning. More specifically, given a set of $n$ physical nodes and communication delays $d(i,j)$ between any two nodes $i,j$, the paper aims to find circulant directed graphs (e.g. ring, exponential, 1-peer exponential) that minimize the maximal delay between any two neighbors in this graph. The proposed method, BTSP-MSR, finds approximate solutions to this combinatorial optimization problem in polynomial time. To do so, the objective is upper bounded by the product of two functions of different variables, yielding two subproblems: the first is the Bottleneck Traveling Salesman Problem (BTSP), for which 2-approximations can be found in polynomial time, the second is a minimization over a set included in $[n]$. Theoretical guarantees on the approximation ratios obtained with BTSP-MSR are provided.

**Compliance With Llm Reviewing Policy:**

Affirmed.

**Final Justification:**

The authors' replies addressed my concerns. I therefore increase my final rating from 3 to 4.

**Key Questions For Authors:**

1.	Theorem 2 would gain from providing explicit forms of $\min_c g(\sigma_c\mathcal{S})$ for specific types of graphs (e.g. exponential, 1-peer exponential). Can such explicit forms be obtained for the graphs you consider?
2.	For Theorem 3 and Assumption 2, is there a point in introducing randomness? Couldn’t the statement be simplified by simply letting $\beta$ be the maximum value of $h(s)$ over every skip width $s$ for given node positions? It is hard to see the point of Theorem 3 and Assumption 2 without concrete examples of distributions for the $X_i$’s, for which $\sqrt{g(\sigma_c \mathcal{S})+\beta}$ is significantly smaller than $g(\sigma_c \mathcal{S})$.
3.	For the MSR step, the minimization is restricted to multiplicative perturbations, in order to preserve the circulant structure. Are there other simple families of permutations that preserve the circulant structure, are small enough to be easily optimized over, and could provide lower MSR factors? In particular, since the current complexity of MSR is $O(n)$ and the cost of BTSP is in $O(n^2\log(n))$, it seems that larger families of permutations could be considered at a negligible computational cost.

Minor comments:
•	L208, eq (4): no $s$ in the maximization problem
•	L229: the inequality is in a smaller font than the rest of the paper, please change this back to 11pt
•	At L753 the font size changes again, affecting the remainder of the supplemental.

**Limitations:**

yes

**Strengths And Weaknesses:**

Strengths:
1.	The problem considered is novel and is well motivated by communication efficiency in decentralized learning.
2.	The trick of exploiting the circulant structures of the considered graphs to simplify the problem down to finding approximate solutions to a BTSP is an interesting idea.
3.	The experiments are convincing, comparing BTSP-MSR to naive baselines on both the BCD minimization problem and their impact in practical decentralized optimization settings. The practical benefits of BTSP-MSR in comparison to other methods is evidenced by the experiments
4.	The improvement of the approximation error in Theorem 3 under the assumption that the delays are an affine function of the Euclidian distance between physical nodes is interesting, and the experiments (e.g. Figure 13) suggest that this bound is much tighter than the one of Theorem 2 in some cases.

Weaknesses:
1.	The theoretical guarantees provided by Theorem 2 seem quite weak, as $\min_c g(\sigma_c \mathcal{S})$ could be of order $n$. While the upper bound at line 235 allows the authors to make the BTSP term appear, which in turns simplifies the procedure by only requiring to find an approximate solution of the BTSP, it also appears to be very loose, leading to possibly poor approximation ratios.
2.	Many quantities are hard to interpret (e.g. $\min_c g(\sigma_c \mathcal{S})$, $\beta, \delta$), and no orders of magnitudes for these quantities are given for motivating examples (e.g. 1-peer exponential graphs), making it difficult to evaluate the strength of the guarantees provided by Theorems 2 and 3.
3.	While some points in Figure 5 correspond to this, it would have been interesting to experimentally assess the sub optimality of BTSP-MSR on small instances for which exact solutions can be found quickly using SCIP.

---

> ### Author Rebuttal · Authors · 2026-03-31
>
> We sincerely thank you for your careful review and valuable feedback. Due to the space limitation, we could not address all questions in full detail. We would be happy to provide further clarification/materials if given the opportunity. __We hope that our responses have addressed your main concerns, and we would greatly appreciate your consideration in updating your evaluation if appropriate.__
>
> # W1: Weak theoretical guarantees in Theorem 2.
>
> First, we would like to clarify a point that may have caused some confusion regarding Theorems 2 and 3 in Sec. 4.3:
> - Theorem 2 provides the __conservative approximation ratio without requiring assumptions__.
> - Theorem 3 derives a __tighter approximation ratio under several assumptions__ that can hold in practical NW scenarios. In Assumption 1, $d$ is designed based on Euclidean distance, and the physical nodes are assumed to be geographically randomly distributed (Assumption 2).
>
> __We would like you to understand that Theorem 2 is intended to provide a conservative approximation ratio; thus, some looseness in the bound is natural.__ In deriving Theorem 2, we found that this looseness mainly arises from the inequality $d(\pi(i), \pi([i+s])) \le sB(\pi)$. Moreover, this inequality becomes tight only when the physical nodes are arranged on a straight line, although such a configuration is not realistic in practice. On the other hand, in Theorem 3, we succeeded in providing a tighter approximation ratio by incorporating that (A1) communication delays can be modeled by Euclidean distance and (A2) nodes are geographically randomly distributed.
>
> # W2: Strength of the guarantees provided by Theorems 3.
>
> We first clarify the three quantities in Theorem 3:
>
> - $h(s)$: the maximum correlation in communication directions in any consecutive $s+1$ physical nodes along the BTSP tour.
> - $\beta$: a high-probability upper bound on $h(s)$.
> -  $\delta$: the probability of failure in the high-probability analysis.
>
> Among these, $h(s)$ may be difficult to interpret. To provide intuition, we consider three scenarios of node positions:
>
> __(Scenario 1) Aligned in a straight line.__
> This corresponds to the most conservative scenario discussed in (W1), where nodes $\pi(i), \ldots, \pi([i+s])$ are aligned in a straight line. Although $h(s)$ can be computed analytically as $\max_s h(s) = O(n^2)$; however, it yields the most conservative approximation ratio in Theorem 2.
>
> __(Scenario 2) Placed on concentric circles.__
> $h(s)$ can be analytically tractable in this case. When $n = 4k^2$ nodes are placed evenly on $2k$ concentric circles, the BTSP route alternates between inner and outer circles, and the distance between adjacent nodes is $\Theta(1/k)$. In this case, $h(s) \le \max_i \left\| \sum_{t=0}^{s-1} e_{[i+t]} \right\|^2 = O(k^2) = O(n).$
>
> __(Scenario 3) Uniform random distribution.__
> We consider this setting close to practical NW scenarios. Although $h(s)$ is difficult to evaluate analytically, it can be estimated empirically from randomly generated positions, yielding a high-probability bound $\beta$ (Appendix E.3). Using this estimate, Figure 13 shows that the practical upper bound in Theorem 3 is significantly tighter than that of Theorem 2.
>
> # W3: Investigation of suboptimality of BTSP-MSR solutions in small instances.
>
> As suggested, we evaluated the suboptimality of BTSP-MSR using small instances ($10 \le n \le 15$), and the results are illustrated in the following figure: __https://anonymous.4open.science/r/ICML2026_rebuttal-F654/opt_proposed_bound.png__
>
> The results show that __BTSP-MSR consistently achieves near-optimal solutions across all tested NW topologies, at least for the small values of $n$ considered__.
>
> # Q2: Why introducing randomness in Theorem 3 and Assumption 2?
>
> First, as you pointed out, we can derive an upper bound using $\max_s h(s)$ for a given node positions and communication delay. For example, for the Ebone NW, we can estimate the tighter upper bound as discussed in Appendix F.4. Further, we can use $\max_s h(s)$ instead of $\beta$, and we have:
> - $\max_s h(s) = 1.12$,
> -  Theorem 2: $\min_c g(\sigma_c \mathcal{S})=31$,
> -  Theorem 3: $\sqrt{\min_c g(\sigma_c \mathcal{S}) + \max_s h(s)} = 5.67$,
>
> which gives a concrete example that Theorem 3 provides a tighter bound than Theorem 2.
>
> A more concrete example is (Scenario 2) in (W2), for which $\max_s h(s)$ can be evaluated analytically. In this case, $\max_s h(s) = O(n)$, implying that for large nodes size $n$, $\sqrt{\min_c g(\sigma_c \mathcal{S}) + \max_s h(s)}$ becomes significantly smaller than $g(\sigma_c \mathcal{S})$.
>
> That said, we introduce randomness in Theorem 3 to avoid rare but adversarial node positions that would otherwise dominate the bound. As shown in Figure 12, small $\delta$ leads to large estimates of $\beta$, reflecting these adversarial events. The high-probability formulation in Theorem 3 effectively excludes such statistically unlikely patterns, resulting in a much tighter bound.

---

> > ### Author Rebuttal · Reviewer_7MoZ · 2026-04-03
> >
> > The answers and additional experiments address part of my concerns. Due to a lack of space, the authors were not able to answer my third question, I would appreciate if they could provide an answer to it.

---

> > > ### Author Response · Authors · 2026-04-06
> > >
> > > We sincerely appreciate your follow-up and the opportunity to address the remaining concerns.
> > > # Q1: Explicit form of approximation ratios for specific types of graphs (e.g., exponential, 1-peer exponential).
> > >
> > > We understand that Q1 is related to W1, namely that "the theoretical guarantees provided by Theorem 2 seem quite weak, as $\min_c g(\sigma_c \mathcal{S})$ could be of $O(n)$", and that you are asking whether a more precise analysis can be obtained for specific graphs.
> > >
> > > First, recall that MSR term is "$g(\mathcal{S}) = \frac{1}{\tau}\sum_{l=0}^{\tau-1} \max S^{(l)}$" (Lemma 2 in Sec. 4.1(b)); with the multiplicative permutation $\sigma_c$, it can be rewritten by
> > > $$ g(\sigma_c \mathcal{S}) = \frac{1}{\tau}\sum_{l=0}^{\tau-1} \max_{s \in S^{(l)}} [cs],$$
> > > where $[cs] = cs \bmod n$.
> > > "Furthermore, using the symmetry of $d$, we can reduce $g(\sigma_c S) \le \lfloor n/2 \rfloor$ (Lemma 3 in Appendix C)" (see line 243--244 in Sec. 4.2 (c)).
> > > This implies that, in the worst case, $\min_c g(\sigma_c \mathcal{S}) = O(n)$, as in (W1).
> > >
> > > Next, as you suggested, __we derived the MSR term for exponential and 1-peer exponential graphs to investigate whether the MSR term can be reduced.__
> > >
> > > __Exponential graph.__
> > > The MSR term for the exponential graph is given by $g(\sigma_c \mathcal{S}) = \max_{s} \min\{[cs], n-[cs]\}$. However, due to the presence of the modulo operation, we were unable to derive a simple closed-form expression.
> > >
> > > Instead, __we show that $\min_c g(\sigma_c \mathcal{S}) = \Theta(n)$ with limited $n$, indicating that the asymptotic order of the MSR term is unchanged, while leaving open the possibility of improvement in the constant level.__
> > >
> > > __Lemma (Lower bound on the MSR term for the exponential graph with limited $n$).__
> > >
> > > When $n$ is limited to $n=2^k-1$ with an integer $k\ge 2$,
> > > the lower bound on the MSR term for the exponential graph holds:
> > > $$\min_c g(\sigma_c \mathcal{S}) \ge \frac{n+1}{4}.$$
> > > __Proof sketch.__ (Detailed proof will be included in the revised version. )
> > > Our key idea is that for $n=2^k-1$, $[c2^l]$ corresponds to a cyclic shift of the $k$-bit binary representation of $c$.  Hence, $\sigma_c \mathcal{S}$ consists of all cyclic shift of $c$. Since $0 < c < 2^k-1$, its binary representation is neither all-zero nor all-one. Thus, there exists a cyclic shift whose first two bits are $01$ in $\sigma_c \mathcal{S}$. Such a value is at least $2^{k-2}=(n+1)/4$, yielding the lower bound.
> > >
> > > __In summary, for the exponential graph, $\min_c g(\sigma_c \mathcal{S}) = \Theta(n)$, while the constant factor may be improved.__
> > >
> > > __1-peer exponential graph.__ The MSR term $g(\sigma_c \mathcal{S}) = \frac{1}{\tau} \sum_{s} \min\{[cs], n-[cs]\}$ is an average over $\lceil \log_2 n \rceil$ terms, each bounded by $O(n)$. Since $\tau = \lceil \log_2 n\rceil + 1 = O(\log n)$, we obtain
> > > $$
> > > \min_c g(\sigma_c \mathcal{S}) = O\left(\frac{n}{\log n}\right),
> > > $$
> > > __which is tighter than $O(n)$, suggesting that the MSR term can be reduced for the 1-peer exponential graph.__
> > >
> > > # Q3: Other simple families of permutations for MSR step.
> > > Multiplicative permutations are the most fundamental and simple choice, since they are efficiently enumerable in $O(n)$ and applicable to arbitrary node sizes $n$.
> > > However, as you insightfully pointed out in your question—"Are there other simple families of permutations that preserve the circulant structure [...]? [...] larger families of permutations could be considered at a negligible computational cost"—this is indeed a reasonable direction.
> > >
> > > During the rebuttal period, we examined other structure-preserving permutations. __When $n$ is square-free (i.e., a product of distinct primes), only multiplicative permutations preserve the circulant structure (Muzychuk, 1995).__
> > >
> > > When $n$ is not square-free, there may be other permutations preserving the given circulant structure, which may reduce the MSR. For example, consider $n=16$, then the circulant digraphs generated by $S_1 = \\{1, 2, 7\\}$ and $S_2 = \\{2, 3, 5\\}$ have the same circulant structure, but their equivalence cannot be realized by any multiplicative permutation. Since $g(S_1) = 7 > g(S_2) = 5$, the MSR can be reduced from $7$ to $5$ for $S_1$. Namely, __when $n$ is not square-free, the MSR may be reduced by a non-multiplicative permutations.__
> > >
> > > From this observation, we additionally searched for non-multiplicative permutations that further reduce the MSR, focusing on exponential, 1-peer exponential, and sparse exponential graphs, with the test settings in Sec. 5: (T1) with $n \in [8, 128]$, including (T2) with $n=87$; but found none. This suggests that, __in (T1) and (T2) in Sec. 5, multiplicative permutations already capture all structure-preserving permutations.__
> > >
> > > Since these discussions are informative, we will include them in the revised version.
> > > We sincerely thank you for giving us the opportunity to consider these important points.
> > >
> > > (Muzychuk, 1995): Muzychuk, Adám's Conjecture Is True in the Square-Free Case, 1995.

---

### Official Review · Reviewer_mh56 · 2026-03-12

**Soundness:** 3
**Presentation:** 3
**Significance:** 2
**Originality:** 3
**Overall Recommendation:** 4
**Confidence:** 3

**Summary:**

This paper addresses the efficiency bottleneck problem faced by decentralized learning in heterogeneous communication delay environments and proposes a node allocation approximate solution method named BTSP-MSR. The authors explore the structural characteristics of the cyclic directed graph and point out that any cyclic directed graph can be regarded as the union of multiple directed ring graphs. Based on this discovery, the article derives the theoretical upper bound of BCD and decomposes it into two manageable terms. The BTSP-MSR algorithm minimizes these two terms in a two-stage strategy to efficiently obtain an approximately optimal node allocation scheme. At the theoretical level, the paper not only establishes the approximation ratio of the algorithm in the worst case but also derives a tighter approximation ratio under assumptions that conform to actual physical network scenarios. At the experimental level, numerical evaluations show that in large-scale node configurations, BTSP-MSR can stably and significantly reduce the BCD of various cyclic directed graphs.

**Compliance With Llm Reviewing Policy:**

Affirmed.

**Final Justification:**

The theory of this article is rigorous and has certain practical significance. Therefore, I maintained my original positive score.

**Key Questions For Authors:**

1. In real decentralized learning, compared with communication delay, communication bandwidth and computing speed are important reasons for increasing physical delay and system bottleneck. Although not discussed in this paper, whether the author's work can be compatible with the joint optimization of these contents, no additional content is needed, please briefly discuss this possibility.
2. Is it a common process to create a perfect virtual topology and then map it to the actual physical network nodes? In the design of distributed cluster based on physical topology awareness, the bandwidth and delay are usually determined first, and then the optimal communication topology is dynamically determined. Could the author give examples or cite existing research to explain the existence of requirements?

**Limitations:**

see weakness

**Strengths And Weaknesses:**

Strengths
1. The article initially addressed the node allocation problem for advanced network topologies (such as exponential graphs, 1-peer exponential graphs, etc., which are sparse structures) under heterogeneous communication delays. The author utilized the mathematical property that "cyclic directed graphs can be decomposed into a union of ring graphs", thereby reducing the originally computationally exponential (of factorial order) NP-hard problem to two manageable components.
2. The proposed BTSP-MSR approximation algorithm has a time complexity of only O(n² log n). In numerical experiments, even when dealing with large-scale physical networks consisting of up to 2000 computing nodes, this algorithm can quickly complete the solution.
3. This paper not only derives the worst-case theoretical approximation ratio bound (Theorem 2), but also derives a tighter probability upper bound (Theorem 3) for the characteristics of real-world physical networks, such as Euclidean distance based delay and limited edge direction correlation.
4. In the experiments of real physical network and simulated environment, the proposed algorithm is not only superior to the comparison algorithms, but also effectively improves the overall test accuracy and convergence speed of decentralized learning.

Weaknesses
1. To run the algorithm, the system must obtain the global and complete communication delay matrix between all physical nodes in advance (that is, the network probe data at the level of $O(n^2)$), which leads to huge communication overhead
2. This paper focuses on the statically optimal allocation scheme, a statically locked strategy that appears relatively backward in modern algorithmic frameworks. Modern decentralized learning also increasingly prefers to use sparse graphs (time-varying topologies) that dynamically change over Time, rather than fixing a complex topology.

---

> ### Author Rebuttal · Authors · 2026-03-31
>
> Thank you for your thoughtful review. We hope that our responses have addressed your concerns. Should you have any further questions, we would be more than happy to provide additional clarification. Thank you again for your time and support.
>
> # W1: Communication overhead to obtain the communication delay.
>
> As you stated, BTSP-MSR requires access to the complete communication delay matrix between all physical nodes in advance, which may incur additional communication overhead. However, BTSP-MSR is intended to be executed __only once prior to the start of decentralized learning__ (see Figure 9 in Appendix A), rather than being used repeatedly during training. Therefore, the associated overhead is incurred only once and does not affect the iterative learning process. In addition, the computational complexity of BTSP-MSR is $O(n^2 \\log n)$, and this order remains unchanged even when accounting for the cost of obtaining the communication delay matrix.
>
> # W2: Concern about time-varying topologies.
>
> As you pointed out, sparse graphs with time-varying structures have been explored, although the amount of work is limited. For example, MATCHA [Wang et al., 2022], which is also evaluated in Sec. 5, constructs a randomized sparse graph with a preference for geographically neighboring nodes. MATCHA performs 1-peer communication at each round and focuses on reducing communication delay. However, such methods typically result in relatively small spectral gaps, which can limit their effectiveness in decentralized learning. In our experiments (T2: Figure 8 in Sec. 5), MATCHA underperforms compared to methods based on advanced NW topologies. In contrast, the experiments suggest that combining advanced NW topologies with large spectral gaps (e.g., exponential or 1-peer exponential graphs) and BCD reduction can lead to more effective decentralized learning (see also our response [W1] to Reviewer EBs7).
>
> Our work is motivated by the relative lack of practical approaches for reducing communication delay, despite extensive research on advanced NW topologies in decentralized learning. That said, we agree that dynamically optimizing both NW topology and delay is an interesting direction, and we will highlight it as a potential future work in the revised version.
>
> # Q1: Importance of communication bandwidth and computing speed.
>
> In this work, we define the distance function $d$ as the communication delay, following prior work (e.g., [Marfoq et al., 2020]). However, by extending the definition of $d$ to incorporate additional factors such as communication bandwidth and computing speed, it would be possible to address the joint optimization of these aspects, as you suggested. We will include a discussion of this direction in the revised version.
>
> # Q2: Validity of fixing virtual topology.
>
> We would like to emphasize that we follow the prior work [Marfoq et al., 2020] to formulate the problem.
> As discussed in our response W1 to Reviewer EBs7, analytical estimates of total runtime based on convergence rates suggest that combining advanced NW topologies with large spectral gaps and BCD reduction is likely to improve overall efficiency. Since advanced NW topologies have been extensively studied in decentralized learning, we believe it is natural to investigate how to effectively deploy them in practical environments by reducing BCD.
>
> On the other hand, MATCHA [Wang et al., 2022] is an approach closer to your suggestion, which dynamically constructs sparse communication NWs based on communication delay. However, as shown in our experiments (T2: Figure 8 in Sec. 5), the advanced NW topologies with BTSP-MSR consistently outperform MATCHA. This is likely because, while MATCHA effectively reduces BCD, it results in relatively small spectral gaps, which limit the convergence rate.
>
> We agree that jointly optimizing both spectral gap and BCD is a promising direction, and we will include a discussion of this in the revised version.

---

> > ### Author Rebuttal · Reviewer_mh56 · 2026-04-02
> >
> > I appreciate the authors' response, and all of my questions have been fully addressed. I will maintain my original positive score.

---

> > > ### Author Response · Authors · 2026-04-02
> > >
> > > Thank you again for your constructive review. We would be happy to address any additional questions if they arise.

---

### Official Review · Reviewer_cDYa · 2026-03-13

**Soundness:** 3
**Presentation:** 3
**Significance:** 3
**Originality:** 3
**Overall Recommendation:** 5
**Confidence:** 2

**Summary:**

The paper studies the problem of efficient implementation of virtual decentralized communication protocols on physical communication network. The problem is formalized as minimization of maximum (over pairs of communicating nodes) time to transmit the message over all possible mappings between nodes of virtual communication graph to nodes of physical communication network. An approximation algorithm is proposed with theoretical and experimental analysis.

**Compliance With Llm Reviewing Policy:**

Affirmed.

**Key Questions For Authors:**

- It seems resonable to add a heuristic combinatorial optimization algorithm such as simulated annealing to baselines
- Spectral gaps in Figure 7 were estimated by the empirical convergence rate of the consensus algorithm. What is the formal definition of spectral gap for dynamic topologies? Isn't it possible to compute its exact value analytically or numerically as for static topologies(weight matrices)?

**Limitations:**

Yes

**Strengths And Weaknesses:**

The paper is well written and the authors provide extensive theoretical and numerical analysis.
According to the authors, the problem of node assignemnt for communication schemes in this generality (virtual communication graph is a circulant graph) were not previously addressed. However applying predefined communication scheme to a given physical network seems suboptimal comparing to designing it from scratch.

---

> ### Author Rebuttal · Authors · 2026-03-31
>
> We sincerely appreciate the time and expertise you have devoted to reviewing our paper. We hope that our responses have addressed your concerns. Should you have any further questions, we would be more than happy to provide additional clarification. Thank you again for your time and support.
>
> # W1: Applying predefined communication scheme to a given physical network seems suboptimal comparing to designing it from scratch.
>
> As you recognized, our strategy is based on leveraging advanced NW topologies with large spectral gaps (e.g., exponential graph and 1-peer exponential graph), and reducing BCD by fixing their topologies. We agree that this strategy may be suboptimal, in the sense that jointly optimizing both the NW topology and BCD from scratch could potentially lead to better performance. We will highlight this direction as a future research direction in the revised version.
>
> # Q1: Comparison with simulated annealing.
>
> We conducted additional experiments in (T1) comparing our method with simulated annealing. Since our paper particularly focuses on the effectiveness of BTSP-MSR in large-scale settings, we evaluated the methods for $n\in \\{64, \cdots, 128\\}$. Please see the corresponding figure:
> __https://anonymous.4open.science/r/ICML2026_rebuttal-F654/sa_vs_proposed.png__
>
> From the results, we observe that BTSP-MSR outperforms simulated annealing in most cases, especially for larger numbers of nodes $n$. In the revised version, we will include this discussion.
>
> # Q2: Formal definition of spectral gap for dynamic topologies.
>
> As noted in the footnote on p. 2, for dynamic NW topologies, __the spectral gap can be defined as that of the expected product of the mixing matrices applied across multiple communication rounds__ (see Assumption 3 in Appendix B.3):
>
> $$
> E_W \bigl[ ||X W - \bar{X} ||_F^2\bigr] \le (1 - \lambda)|| X - \bar{X} ||_F^2,
> $$
>
> where $W=W^{(0)} \cdots W^{(\tau-1)}$ denotes the product of mixing matrices across multiple communication rounds. For structurally regular NW topologies such as the 1-peer exponential graph, it is possible to derive the spectral gap analytically (see Table 1 in Appendix B.2).
>
> On the other hand, in Figure 7 (Sec. 5), we report empirically estimated spectral gaps per round, rather than analytical values. This is because, for random dynamic graphs, especially those with non-uniform connection distributions such as MATCHA (Wang et al., 2022), it is generally difficult to derive the spectral gap analytically under the above expectation-based definition.
>
> To ensure a fair comparison across different NW topologies, we therefore estimate the spectral gap from experimental results. The estimation procedure is described in Appendix F.3.

---

> > ### Author Rebuttal · Reviewer_cDYa · 2026-04-03
> >
> > W1: The authors agreed that joint optimization of both metrics is better, but more difficult. Highlighting this as a future research direction would suffice.
> >
> > Q1: Thank you for conducting additional experiments! It is not clear now whether SA underperformed due to insufficient tuning of its parameters (temperature schedule, candidate selection method). I hope all the details will appear in the updated version of the manuscript. Anyway, I don't think that making a thorough numerical comparison is essential for the paper, so it is ok
> >
> > Q2: Since assumption 3 is not referenced anywhere except B.3 (not referenced in the footnote, B.2, or F.3), it is somewhat confusing. Please add links to the definition of the spectral gap in relevant places to increase readability.

---

> > > ### Author Response · Authors · 2026-04-06
> > >
> > > Thank you for your positive evaluation and for confirming that the concerns have been fully addressed. We would be happy to clarify any remaining points if needed.

---

### Decision · Program_Chairs · 2026-04-30

**Decision:**

Accept (regular)

**Comment:**

This paper proposes a strategy to design (a time varying) communication topology for communication efficient decentralized optimization on physical networks, through formulating the design problem as the minimization of the max transmission time between nodes. The proposed approximation relies on the observation that circulant digraph can be decomposed into directed ring graphs, and subsequently solved as a multi-objective problem involving BTSP and MSR. Although the reviewers found the experimental results to be convincing, there are concerns over the weak/loose theoretical bounds. The concerns are addressed during the rebuttal and the reviewers are in consensus to accept the paper. The authors are reminded to incorporate comments from the reviews in preparing the final version.